



# Implementation of a sigma coordinate system in PALM-Sigma v1.0 (based on PALM v21.10) for LES study of the marine atmospheric boundary layer

Xu Ning[1] and Mostafa Bakhoday-Paskyabi[1]

[1]Geophysical Institute, University of Bergen and Bergen Offshore Wind Centre, Allégaten 55, 5007 Bergen, Norway

**Correspondence:** Xu Ning (xu.ning@uib.no)

**Abstract.** In large-eddy simulation studies of the marine atmospheric boundary layer, wind–wave interactions are often over-simplified using wall-stress models parameterized by roughness length, overlooking the complex coupling dynamics, especially under wind–wave non-equilibrium states. Here, we develop a wave-phase-resolved solver that employs a surface-following sigma coordinate system to explicitly resolve evolving wave geometry. Simulations under low-wind conditions with different
wave regimes reproduce characteristic features of wave-driven winds reported in previous studies. Notably, the results show that wave-induced form stress significantly modulates vertical momentum flux, with effects extending well beyond the wave boundary layer. Built on PALM's framework, the solver can be seamlessly integrated with the multi-scale nesting module and coupled with wave models, enabling high-fidelity simulations to improve understanding and parameterization of wind–wave coupling under realistic conditions.

## 1  Introduction

The ocean surface is characterized by ubiquitous waves with a wide range of amplitudes and propagating speeds. These waves induce perturbations to the velocity and pressure fields of the overlying atmospheric boundary layer flow and modify the exchange of mass, momentum, and heat between the atmosphere and the ocean (Sullivan and McWilliams, 2010). Although the wave-induced disturbances are known to decay with height by an exponential rate (Donelan et al., 2006), numerous studies

demonstrate that the surface waves have profound influences on the wind field above (Peña and Gryning, 2008; Grare et al., 2018; Chen et al., 2019), especially in conditions with the presence of swells under relatively low wind speed (Grachev and Fairall, 2001; Hanley and Belcher, 2008). A deep understanding of this atmosphere-wave interaction process is thus beneficial for the study of marine meteorology and relevant applications from weather forecasting to offshore wind energy. However, this problem remains a challenging topic in both theoretical and experimental researches due to the complexity of physics

and demanding requirements for the measurement devices. More work is necessary to obtain a comprehensive knowledge of the wave-induced effects on the marine atmospheric boundary layer (MABL). Over the past two decades, the numerical modeling tools, especially the large-eddy simulation technique, have experienced rapid development and been widely used in the area of micro-scale meteorology. The latest high-performance computing (HPC) equipment enables numerical simulations with a magnitude of grid points up to $10^{10}$ (Kröniger et al., 2018), about 10 thousands times larger than 20 years before. The



ability to resolve 3D flow field with high temporal-spatial resolution makes large-eddy simulation a promising tool to study the wave-coherent flow in the vicinity of the wave surface and its effects on the boundary layer structure.

In numerical modeling of the marine atmospheric boundary layer flows, wave effects are typically incorporated using two main approaches (Deskos et al., 2021). The first is a wave-phase-averaged method, where wave-induced stress near the surface is parameterized using empirical formulations based on wind-wave conditions. The second is a wave-phase-resolved approach,

in which the geometry of the wavy sea surface is explicitly represented at the bottom boundary, allowing the flow disturbances generated by surface motions to be directly resolved. Considering its simplicity and low cost, the first approach is used in most of the current simulation tools. Nevertheless, the reliability of parameterizations shows dependence on site and wave types (Drennan et al., 2005; Jiménez and Dudhia, 2018), and there is no universal method with satisfying performance in any wind-sea environments. By contrast, the second approach has a more accurate representation of the physics of the airflow-water

interaction process by involving the information of wave motions as the boundary condition. This enables the wave-phase-resolved simulator to reproduce the coupling dynamics in various wind-sea regimes and stability conditions, e.g., wind-wave misalignment and upwards wave-induced momentum flux in swell conditions (Smedman et al., 1999, 2009). Therefore, many research groups have developed numerical simulation tools with wave-phase-resolving capability in order to study the dynamics of turbulent flow affected by different types of waves.

The direct numerical simulation (DNS) studies by Sullivan et al. (2000); Sullivan and McWilliams (2002); Shen et al. (2003); Kihara et al. (2007); Yang and Shen (2009, 2010) provided detailed flow field information in the vicinity of water surface and valuable interpretations of the process and mechanism of wind-wave interaction. Such direct numerical simulations must be confined to a low Reynolds number due to the high computational cost for resolving the wall boundary layer flow. Compared with DNS, LES has no such limitation if a wall-layer model is applied to estimate the surface stress by the Monin-Obukhov

similarity theory (MOST). This makes the LES model a powerful tool to study the wave-induced effects on MABL in more realistic scenarios. Sullivan et al. (2008, 2014) considered monochromatic and broadband-spectral waves and a wide range of geostrophic wind speeds in his LES study. His simulations illustrated that both the mean wind profile and the turbulence in MABL can be strongly affected by waves. The balance between the turbulent friction, wave-induced form stress, and subgrid-scale (SGS) stress depends on the wave age (the ratio between wave phase speed and friction velocity) and the distance above

the water surface. Hao and Shen (2019) performed a two-way wind-wave coupling simulation using LES and potential flow solver to solve wind turbulence and wave field, respectively. They found the phenomenon of the frequency shift in the wind-wave spectrum and the enhanced turbulent shear stress from sweep events as a result of waves. In addition, as the fast growing offshore wind energy industry draws increasing attention over the world, the wave-phase-resolved LES technique has also been used in the studies of the wake dynamics of wind turbines in the MABL. For example, Yang et al. (2014) investigated the wind

farm performance under fetch-limited and fully developed wave conditions and indicated that the latter situation with lower surface drag leads to an increase of $8\%$ for the power extraction rate. The recent series of LES studies by Yang et al. (2022b, a) were focused on swell-dominant regimes. They found that the power output of a wind turbine can increase by nearly one time under the influence of fast propagating wind-following swells, and the recovery of wake flow is also affected through enhanced or mitigated turbulence due to the presence of waves.





The above numerical work contributes greatly to our understanding of how the characteristics of wind field change while interacting with the underlying wave surface, but there are still knowledge gaps in wind-wave coupling problem with more complex environmental conditions, such as non-neutral atmospheric stability and the wake flow of large-scale wind farms. This becomes the main motivation for our present work to extend the PALM (Parallelized Large-Eddy Model) code with the ability to solve the turbulent flow over wavy surface. PALM is a widely used and well maintained code developed by the group from Leibniz Universitat Hannover (LUH) (Maronga et al., 2020) since 20 years ago. It is designed for highly parallelized simulations of atmospheric and oceanic boundary layer flows and is embedded with many practical sub-models, such as meso-to-micro scale nesting interface, Lagrangian particle model, and wind turbine model, which makes PALM a powerful tool in diverse research fields. The current work further develop PALM to a wave-phase-resolved simulator (PALM-Sigma) through the implementation of a sigma coordinate system and pave the way for its application in future studies related to MABL flow and offshore wind energy engineering. We first present the governing equations and their numerical implementation in Sects. 2 and 3, followed by an overview of the code framework in Sect. 4. Section 5 describes the simulation setup and case configurations. The results are discussed in detail in Sect. 6, and finally, Sect. 7 provides a summary and outlines directions for future work.

## 2 Governing equations

### 2.1 Governing equations in Cartesian coordinates

We briefly summarize the governing equations of the original PALM model system in this subsection as the foundation for our subsequent implementation of sigma coordinates. PALM is a large-eddy simulation model that solves the incompressible Navier-Stokes equations. Specifically, the equations consist of the continuity equation, the momentum equation, and the scalar transport equation:

$$\frac{\partial u_j}{\partial x_j} = 0, \tag{1}$$

$$\frac{\partial u_i}{\partial t} = -\frac{\partial u_i u_j}{\partial x_j} - \varepsilon_{ijk} f_j u_k + \varepsilon_{i3j} f_3 u_{g,j} - \frac{1}{\rho_0}\frac{\partial \pi^*}{\partial x_i} + g\frac{\theta_v - \langle\theta_v\rangle}{\langle\theta_v\rangle}\delta_{i3} - \frac{\partial}{\partial x_j}\left(\overline{u_i'' u_j''} - \frac{2}{3}e\delta_{ij}\right), \tag{2}$$

$$\frac{\partial \phi}{\partial t} = -\frac{\partial u_j \phi}{\partial x_j} - \frac{\partial \overline{u_j'' \phi''}}{\partial x_j} + \Psi_\phi. \tag{3}$$

Symbols $\epsilon$ and $\delta$ represent the Levi-Civita tensor and Kronecker delta, respectively. $t$ is time, and $x_i$, $x_j$, $x_k$ denote spatial coordinates, with indices $i, j, k \in 1, 2, 3$. $u_i$, $u_j$, and $u_k$ denote the components of velocity. $\pi^*$ represents the modified perturbation pressure, and $\theta_v$ stands for the potential temperature, and $\langle\theta_v\rangle$ is its horizontal average. $\phi$ represents a generic transported scalar, and $\Psi_\phi$ is its source term. These prognostic variables are resolved-scale quantities, with the filtering notation omitted





for clarity. $g$ is the gravitational acceleration and $f$ denotes the Coriolis parameter. $\rho_0$ is the density of dry air. The overbar
denotes Reynolds averaging, and the double prime indicates subgrid-scale (SGS) components. $e$ represents the subgrid-scale
turbulent kinetic energy (SGS-TKE).

## 2.2 Coordinate transformation

The wavy ocean surface introduces a dynamic and curvy lower boundary to the marine atmospheric boundary layer, which
poses significant challenges for numerical modeling in a traditional Cartesian coordinate system. To explicitly resolve the
wave geometry and its interaction with the flow, the PALM model has been extended with a sigma coordinate system. This
transformation normalizes the vertical coordinate by the instantaneous local domain height, defined as the distance between
the evolving wave surface and the fixed upper boundary, while retaining Cartesian coordinates in the horizontal directions. The
coordinate transformation is given by:

$$\tau = t, \quad \xi = x, \quad \psi = y, \quad \zeta = \frac{z - \eta}{H - \eta}, \tag{4}$$

where $\eta$ denotes wave surface elevation and $H$ is the total vertical extent of the domain. $\tau$, $\xi$, $\psi$, $\zeta$ (or equivalently $\xi_1 = \xi$,
$\xi_2 = \psi$, $\xi_3 = \zeta$) are the time and space coordinates in the computational domain. The derivative terms in the physical domain
can then be reformulated by the chain rule as

$$\frac{\partial}{\partial t} = \frac{\partial}{\partial \tau} + \zeta_t \frac{\partial}{\partial \zeta}, \quad \frac{\partial}{\partial x} = \frac{\partial}{\partial \xi} + \zeta_x \frac{\partial}{\partial \zeta}, \quad \frac{\partial}{\partial y} = \frac{\partial}{\partial \psi} + \zeta_y \frac{\partial}{\partial \zeta}, \quad \frac{\partial}{\partial z} = \zeta_z \frac{\partial}{\partial \zeta}, \tag{5}$$

$$\frac{\partial^2}{\partial x^2} = \frac{\partial^2}{\partial \xi^2} + 2\zeta_x \frac{\partial^2}{\partial \xi \partial \zeta} + \zeta_x^2 \frac{\partial^2}{\partial \zeta^2} + \zeta_{xx} \frac{\partial}{\partial \zeta}, \quad \frac{\partial^2}{\partial y^2} = \frac{\partial^2}{\partial \psi^2} + 2\zeta_y \frac{\partial^2}{\partial \psi \partial \zeta} + \zeta_y^2 \frac{\partial^2}{\partial \zeta^2} + \zeta_{yy} \frac{\partial}{\partial \zeta}, \quad \frac{\partial^2}{\partial z^2} = \zeta_z^2 \frac{\partial^2}{\partial \zeta^2}. \tag{6}$$

The derivatives of $\zeta$ with respect to the original spatial coordinates, commonly referred to as metric terms (Pletcher et al.,
2012), establish the mathematical relationship between the physical and computational domains. These metric terms are ex-
pressed as

$$\zeta_t = \frac{\zeta - 1}{H - \eta} \eta_t, \quad \zeta_x = \frac{\zeta - 1}{H - \eta} \eta_x, \quad \zeta_y = \frac{\zeta - 1}{H - \eta} \eta_y, \quad \zeta_z = \frac{1}{H - \eta}, \tag{7}$$

$$\zeta_{xx} = \frac{2(\zeta - 1)}{(H - \eta)^2} \eta_x^2 + \frac{\zeta - 1}{H - \eta} \eta_{xx}, \quad \zeta_{yy} = \frac{2(\zeta - 1)}{(H - \eta)^2} \eta_y^2 + \frac{\zeta - 1}{H - \eta} \eta_{yy}. \tag{8}$$

## 2.3 Governing equations in Sigma coordinates

By applying the above mesh transformation to the original governing equations in PALM, we obtain the Navier-Stokes equa-
tions and scalar transport equation in the sigma coordinate system:

$$\frac{\partial u}{\partial \xi} + \zeta_x \frac{\partial u}{\partial \zeta} + \frac{\partial v}{\partial \psi} + \zeta_y \frac{\partial v}{\partial \zeta} + \zeta_z \frac{\partial w}{\partial \zeta} = 0, \tag{9}$$



$$\frac{\partial u}{\partial \tau} + \zeta_t \frac{\partial u}{\partial \zeta} = -\left(\frac{\partial uu}{\partial \xi} + \zeta_x \frac{\partial uu}{\partial \zeta}\right) - \left(\frac{\partial uv}{\partial \psi} + \zeta_y \frac{\partial uv}{\partial \zeta}\right) - \zeta_z \frac{\partial uw}{\partial \zeta} - f_2 w + f_3(v - v_g) - \frac{1}{\rho_0}\left(\frac{\partial \pi^*}{\partial \xi} + \zeta_x \frac{\partial \pi^*}{\partial \zeta}\right)$$
$$- \left(\frac{\partial}{\partial \xi} + \zeta_x \frac{\partial}{\partial \zeta}\right)\left(\overline{u''u''} - \frac{2}{3}e\right) - \left(\frac{\partial}{\partial \psi} + \zeta_y \frac{\partial}{\partial \zeta}\right)(\overline{u''v''}) - \zeta_z \frac{\partial}{\partial \zeta}(\overline{u''w''}), \tag{10}$$

$$\frac{\partial v}{\partial \tau} + \zeta_t \frac{\partial v}{\partial \zeta} = -\left(\frac{\partial vu}{\partial \xi} + \zeta_x \frac{\partial vu}{\partial \zeta}\right) - \left(\frac{\partial vv}{\partial \psi} + \zeta_y \frac{\partial vv}{\partial \zeta}\right) - \zeta_z \frac{\partial vw}{\partial \zeta} - f_3(u - u_g) - \frac{1}{\rho_0}\left(\frac{\partial \pi^*}{\partial \psi} + \zeta_y \frac{\partial \pi^*}{\partial \zeta}\right)$$
$$- \left(\frac{\partial}{\partial \xi} + \zeta_x \frac{\partial}{\partial \zeta}\right)(\overline{v''u''}) - \left(\frac{\partial}{\partial \psi} + \zeta_y \frac{\partial}{\partial \zeta}\right)\left(\overline{v''v''} - \frac{2}{3}e\right) - \zeta_z \frac{\partial}{\partial \zeta}(\overline{v''w''}), \tag{11}$$


$$\frac{\partial w}{\partial \tau} + \zeta_t \frac{\partial w}{\partial \zeta} = -\left(\frac{\partial wu}{\partial \xi} + \zeta_x \frac{\partial wu}{\partial \zeta}\right) - \left(\frac{\partial wv}{\partial \psi} + \zeta_y \frac{\partial wv}{\partial \zeta}\right) - \zeta_z \frac{\partial ww}{\partial \zeta} + f_2 u + g\frac{\theta_v - \theta_{v,\text{ref}}}{\theta_{v,\text{ref}}} - \frac{1}{\rho_0}\left(\zeta_z \frac{\partial \pi^*}{\partial \zeta}\right)$$
$$- \left(\frac{\partial}{\partial \xi} + \zeta_x \frac{\partial}{\partial \zeta}\right)(\overline{w''u''}) - \left(\frac{\partial}{\partial \psi} + \zeta_y \frac{\partial}{\partial \zeta}\right)(\overline{w''v''}) - \zeta_z \frac{\partial}{\partial \zeta}\left(\overline{w''w''} - \frac{2}{3}e\right), \tag{12}$$

$$\frac{\partial \phi}{\partial \tau} + \zeta_t \frac{\partial \phi}{\partial \zeta} = -\left(\frac{\partial u\phi}{\partial \xi} + \zeta_x \frac{\partial u\phi}{\partial \zeta}\right) - \left(\frac{\partial v\phi}{\partial \psi} + \zeta_y \frac{\partial v\phi}{\partial \zeta}\right) - \zeta_z \frac{\partial w\phi}{\partial \zeta}$$
$$- \left(\frac{\partial}{\partial \xi} + \zeta_x \frac{\partial}{\partial \zeta}\right)(\overline{u''\phi''}) - \left(\frac{\partial}{\partial \psi} + \zeta_y \frac{\partial}{\partial \zeta}\right)(\overline{v''\phi''}) - \zeta_z \frac{\partial}{\partial \zeta}(\overline{w''\phi''}) + \Psi_\phi. \tag{13}$$

As in the original PALM code, the SGS fluxes are modeled using an eddy diffusivity approach, where the flux is proportional
to the gradient of the corresponding variable. However, due to the mesh transformation, additional gradient terms involving the
metric coefficients arise. The resulting expressions for the momentum and scalar fluxes are given by

$$\overline{u_i''u_j''} - \frac{2}{3}e\delta_{ij} = -K_m\left(\frac{\partial \xi_k}{\partial x_j}\frac{\partial u_i}{\partial \xi_k} + \frac{\partial \xi_k}{\partial x_i}\frac{\partial u_j}{\partial \xi_k}\right), \tag{14}$$

$$\overline{u_i''\phi''} = -K_h\left(\frac{\partial \xi_k}{\partial x_i}\frac{\partial \phi}{\partial \xi_k}\right). \tag{15}$$

Multiple options of SGS parameterizations can be used to estimate the eddy diffusivities ($K_m$ and $K_h$), including the modified
Deardorff models (Deardorff, 1980; Moeng and Wyngaard, 1988; Saiki et al., 2000) and the dynamic SGS model (Germano
et al., 1991). These models compute eddy diffusivities based on the local grid resolution and the strength of subgrid-scale
turbulent kinetic energy. The SGS-TKE is obtained by solving the following prognostic equation:

$$\frac{\partial e}{\partial \tau} + \zeta_t \frac{\partial e}{\partial \zeta} = -u\left(\frac{\partial e}{\partial \xi} + \zeta_x \frac{\partial e}{\partial \zeta}\right) - v\left(\frac{\partial e}{\partial \psi} + \zeta_y \frac{\partial e}{\partial \zeta}\right) - w\left(\zeta_z \frac{\partial e}{\partial \zeta}\right)$$
$$- \overline{u_j''u''}\left(\frac{\partial u_j}{\partial \xi} + \zeta_x \frac{\partial u_j}{\partial \zeta}\right) - \overline{u_j''v''}\left(\frac{\partial u_j}{\partial \psi} + \zeta_y \frac{\partial u_j}{\partial \zeta}\right) - \overline{u_j''w''}\left(\zeta_z \frac{\partial u_j}{\partial \zeta}\right)$$
$$- \left(\frac{\partial}{\partial \xi} + \zeta_x \frac{\partial}{\partial \zeta}\right)\left[-2K_m\left(\frac{\partial e}{\partial \xi} + \zeta_x \frac{\partial e}{\partial \zeta}\right)\right] - \left(\frac{\partial}{\partial \psi} + \zeta_y \frac{\partial}{\partial \zeta}\right)\left[-2K_m\left(\frac{\partial e}{\partial \psi} + \zeta_y \frac{\partial e}{\partial \zeta}\right)\right]$$
$$- \zeta_z \frac{\partial}{\partial \zeta}\left[-2K_m\left(\zeta_z \frac{\partial e}{\partial \zeta}\right)\right] + \frac{g}{\theta_{v,0}}\overline{w''\theta_v''} - \epsilon. \tag{16}$$

For clarity, all variables and symbols used in the governing equations are summarized in Table 1.





**Table 1.** List of symbols in the governing equations

| Symbol | Dimension | Description |
|---|---|---|
| $t$ | s | Simulation time in the physical domain |
| $\tau$ | s | Simulation time in the computational domain |
| $H$ | m | Total domain height |
| $x_i$ | m | Coordinate on the Cartesian grid ($x_1 = x$, $x_2 = y$, $x_3 = z$) |
| $\xi_i$ | m | Coordinate on the Sigma-grid ($\xi_1 = \xi$, $\xi_2 = \psi$, $\xi_3 = \zeta$) |
| $\eta$ | m | Surface elevation |
| $u_i$ | m s$^{-1}$ | Velocity components ($u_1 = u$, $u_2 = v$, $u_3 = w$) |
| $\theta_v$ | K | Virtual potential temperature |
| $\pi^*$ | Pa | Modified perturbation pressure |
| $e$ | m$^2$ s$^{-2}$ | SGS-TKE |
| $\phi$ | | Prognostic scalar variable |
| $\Psi_\phi$ | | Source term for scalar $\phi$ |
| $u_{g,i}$ | m s$^{-1}$ | Geostrophic wind components ($u_{g,1} = u_g$, $u_{g,2} = v_g$) |
| $f_i$ | s$^{-1}$ | Coriolis parameter |
| $K_m$ | m$^2$ s$^{-1}$ | Eddy diffusivity of momentum |
| $K_h$ | m$^2$ s$^{-1}$ | Eddy diffusivity of heat |
| $\epsilon$ | m$^2$ s$^{-3}$ | SGS dissipation rate |
| $\rho_0$ | kg m$^{-3}$ | Density of air |
| $\delta$ | | Kronecker delta |
| $\varepsilon$ | | Levi-Civita symbol |

## 3 Numerical methods

### 3.1 Spatial discretization

Based on the original PALM framework, a finite difference method with a staggered variable arrangement is employed for the discretization of the governing equations in the sigma coordinate system. Scalar quantities are located at the cell cen-
ters, whereas velocity components are defined at the corresponding cell faces in their respective directions. The metric terms introduced by the mesh transformation are computed and stored at the cell centers.

The advection terms in the momentum equations are discretized using a flux-conserving formulation based on a fifth-order upwind scheme, following the method of Wicker and Skamarock (2002). In the sigma coordinate system, the advection term in each dimension includes a contribution from the flux gradient along $\zeta$ due to the coordinate transformation. For instance,
the semi-discrete formulation of the scalar advection along the $x$-direction (in the physical domain) can be expressed as

$$\mathcal{A}_x(u\phi) = -\left( \frac{\partial u\phi}{\partial \xi} + \zeta_x \frac{\partial u\phi}{\partial \zeta} \right) \approx -\left( \frac{F_e(u\phi) - F_w(u\phi)}{\Delta \xi} + \zeta_x \frac{F_t(u\phi) - F_b(u\phi)}{\Delta \zeta} \right), \tag{17}$$



where $\Delta\xi$ and $\Delta\zeta$ denote the grid spacings in $\xi$- and $\zeta$-directions, respectively. $F_e$, $F_w$, $F_t$, and $F_b$ represent the fluxes across the east, west, top, and bottom faces of a computational cell, evaluated as

$$
\begin{aligned}
F_e(u\phi) &= \frac{u_e}{60}[37(\phi_{k,i+1} + \phi_{k,i}) - 8(\phi_{k,i+2} + \phi_{k,i-1}) + (\phi_{k,i+3} + \phi_{k,i-2})] \\
&\quad - \frac{|u_e|}{60}[10(\phi_{k,i+1} - \phi_{k,i}) - 5(\phi_{k,i+2} - \phi_{k,i-1}) + (\phi_{k,i+3} - \phi_{k,i-2})],
\end{aligned}
\tag{18}
$$


$$
\begin{aligned}
F_w(u\phi) &= \frac{u_w}{60}[37(\phi_{k,i} + \phi_{k,i-1}) - 8(\phi_{k,i+1} + \phi_{k,i-2}) + (\phi_{k,i+2} + \phi_{k,i-3})] \\
&\quad - \frac{|u_w|}{60}[10(\phi_{k,i} - \phi_{k,i-1}) - 5(\phi_{k,i+1} - \phi_{k,i-2}) + (\phi_{k,i+2} - \phi_{k,i-3})],
\end{aligned}
\tag{19}
$$

$$
\begin{aligned}
F_t(u\phi) &= \frac{u_t}{60}[37(\phi_{k+1,i} + \phi_{k,i}) - 8(\phi_{k+2,i} + \phi_{k-1,i}) + (\phi_{k+3,i} + \phi_{k-2,i})] \\
&\quad - \frac{|u_t|}{60}[10(\phi_{k+1,i} + \phi_{k,i}) - 5(\phi_{k+2,i} + \phi_{k-1,i}) + (\phi_{k+3,i} + \phi_{k-2,i})],
\end{aligned}
\tag{20}
$$

$$
\begin{aligned}
F_b(u\phi) &= \frac{u_b}{60}[37(\phi_{k,i} + \phi_{k-1,i}) - 8(\phi_{k+1,i} + \phi_{k-2,i}) + (\phi_{k+2,i} + \phi_{k-3,i})] \\
&\quad - \frac{|u_b|}{240}[10(\phi_{k,i} + \phi_{k-1,i}) - 5(\phi_{k+1,i} + \phi_{k-2,i}) + (\phi_{k+2,i} + \phi_{k-3,i})].
\end{aligned}
\tag{21}
$$


Here, $k$ and $i$ are the indices of grid points in $\zeta$- and $\xi$-dimensions. The effective advection velocities at the cell surfaces are obtained by

$$
\begin{aligned}
u_e &= u_{k,i+1}, \ u_w = u_{k,i}, \\
u_t &= (u_{k,i} + u_{k,i+1} + u_{k+1,i} + u_{k+1,i+1})/4, \\
u_b &= (u_{k,i} + u_{k,i+1} + u_{k-1,i} + u_{k-1,i+1})/4.
\end{aligned}
\tag{22}
$$

### 3.2 Time advancing

A fractional step method is employed for temporal integration (Kim and Moin, 1985), in conjunction with an explicit third-order Runge–Kutta (RK3) scheme (Williamson, 1980). In each time step, the velocity field is first advanced using the current tendencies, namely, the sum of all right-hand-side terms in the momentum equation, excluding the pressure gradient term. This yields an intermediate (or predicted) velocity field. In the subsequent step, a correction is applied by solving a Poisson equation for pressure, ensuring that the final velocity field satisfies the divergence-free (continuity) condition.

To enhance numerical stability, the pressure correction is applied at each of the three substeps of the RK3 scheme, as recommended by Sullivan et al. (2014). The advancement of a velocity component $u_i$ over one full time step $\Delta t$ is performed



as follows:

$$u_i^0 = u_i^n,$$

$$\hat{u}_i^m = u_i^n + \Delta t \sum_{p=0}^{m-1} \beta_{m,p} \mathcal{P}_{re}[u_i^p, \mathcal{M}(\tau_n + \alpha_p \Delta \tau)],$$

$$u_i^m = \hat{u}_i^m - \gamma_m \mathcal{C}_{orr}[\hat{u}_i^m, \mathcal{M}(\tau_n + \alpha_m \Delta \tau)],$$

$$u_i^{n+1} = u_i^3, \tag{23}$$

where $m \in 1, 2, 3$ denotes the substep index within the RK3 scheme. $\alpha_p$ and $\beta_{m,p}$, $\gamma_m$ are standard Runge-Kutta weights. $\mathcal{P}_{re}$
and $\mathcal{C}_{orr}$ represent the operators for the prognostic tendency function and the pressure correction, respectively. The metric terms $\mathcal{M}$, introduced by the sigma coordinate transformation, are time-dependent and evaluated at each substep accordingly.

## 3.3 Pressure solver

The correction of the preliminary velocity field at each substep is carried out using the pressure gradient, as expressed by:

$$u = \hat{u} - \frac{\Delta \tau}{\rho_0}\left(\frac{\partial \pi^*}{\partial \xi} + \zeta_x \frac{\partial \pi^*}{\partial \zeta}\right),$$
$$v = \hat{v} - \frac{\Delta \tau}{\rho_0}\left(\frac{\partial \pi^*}{\partial \psi} + \zeta_y \frac{\partial \pi^*}{\partial \zeta}\right), \tag{24}$$
$$w = \hat{w} - \frac{\Delta \tau}{\rho_0}\left(\zeta_z \frac{\partial \pi^*}{\partial \zeta}\right).$$

Enforcing the divergence-free condition, i.e., Eq. (9), on the corrected velocity field yields the Poisson equation for pressure:

$$\frac{\partial^2 \pi^*}{\partial \xi^2} + \frac{\partial^2 \pi^*}{\partial \psi^2} + (\zeta_x^2 + \zeta_y^2 + \zeta_z^2)\frac{\partial^2 \pi^*}{\partial \zeta^2} + 2\zeta_x \frac{\partial^2 \pi^*}{\partial \xi \partial \zeta} + 2\zeta_y \frac{\partial^2 \pi^*}{\partial \psi \partial \zeta} + (\zeta_{xx} + \zeta_{yy})\frac{\partial \pi^*}{\partial \zeta}$$

$$= \frac{\rho_0}{\Delta \tau}\left(\frac{\partial \hat{u}}{\partial \xi} + \zeta_x \frac{\partial \hat{u}}{\partial \zeta} + \frac{\partial \hat{v}}{\partial \psi} + \zeta_y \frac{\partial \hat{v}}{\partial \zeta} + \zeta_z \frac{\partial \hat{w}}{\partial \zeta}\right). \tag{25}$$

This pressure equation is discretized on the staggered grid using central differences in both horizontal and vertical directions. This inherently avoids the velocity–pressure decoupling problem and provides a second-order spatial accuracy. A multigrid method is employed to efficiently solve Eq. (25), using Gauss–Seidel iterations as the smoothing (relaxation) method combined
with red–black decomposition. For each solution step, four W-cycle iterations with two smoothing steps per grid level are applied, reducing the velocity divergence to a magnitude on the order of $10^{-5}$.

## 3.4 Surface boundary conditions

### 3.4.1 Wave field

The extended PALM code, incorporating the sigma coordinate framework, enables one-way coupling with surface waves by
prescribing the wave field either analytically or via external input files. This implementation allows for the integration of





complex wave conditions, reconstructed from observational data or generated from empirical wave spectra. In the present study, the wave field is simplified to represent linear deep-water waves, allowing the surface elevation to be expressed by a sinusoidal function:

$$\eta(t, x, y) = A \sin[\omega t + k(x \sin \Phi_w + y \cos \Phi_w)], \tag{26}$$

where $\omega$ is the wave frequency (in radians per second), $k$ is the wave number, and $\Phi_w$ denotes the wave propagation direction. As shown in Eq. (7) and (8), the surface elevation $\eta$ provides the necessary geometric information to compute the metric terms required for constructing the transformed computational mesh.

### 3.4.2 Surface velocity

The velocity boundary condition at the bottom surface is specified as a Dirichlet condition, with the velocity components equal to the wave-induced particle velocities at the surface. For linear deep water waves described by Eq. (26), the surface velocity components are given by

$$u_0 = -\omega \eta \sin \Phi_w,$$

$$v_0 = -\omega \eta \cos \Phi_w. \tag{27}$$

The vertical velocity component, $w_0$, is determined according to the kinematic free-surface boundary condition:

$$w_0 = \frac{\partial \eta}{\partial t} + u_0 \frac{\partial \eta}{\partial x} + v_0 \frac{\partial \eta}{\partial y}. \tag{28}$$

### 3.4.3 Wall-stress

Since the airflow below the first grid level is not explicitly resolved, a wall model is required to estimate the Cartesian momentum flux at the lower boundary, i.e., $\overline{u_i'' u_j''}_0$ for $i, j \in 1, 2, 3$. Notably, the vertical component $\overline{w'' w''}_0$ is non-zero at a wavy surface. The procedure begins with the calculation of the local horizontal relative velocity between the airflow and the wave motion at the first grid level $z_1$:

$$U_r = \sqrt{u_r^2 + v_r^2},$$

$$u_r^2 = \left[ (u_1 - u_0) \cos \varphi_x + (w_1 - w_0) \sin \varphi_x \right]^2,$$

$$v_r^2 = \left[ (v_1 - v_0) \cos \varphi_y + (w_1 - w_0) \sin \varphi_y \right]^2, \tag{29}$$

where the subscript 1 denotes values at the first vertical grid level, and $\varphi_x$ and $\varphi_y$ are the local inclination angles of the wave surface in the $x$- and $y$-directions, respectively. Next, the surface momentum fluxes in the local wave-following coordinate





system are estimated using the constant flux assumption:

$$\overline{u''w''}_{0,\text{local}} = -\left[\frac{\kappa}{\ln(z_1/z_0)}\right]^2 |\overline{U}_r|\overline{u}_r,$$

$$\overline{v''w''}_{0,\text{local}} = -\left[\frac{\kappa}{\ln(z_1/z_0)}\right]^2 |\overline{U}_r|\overline{v}_r. \tag{30}$$

where $\kappa$ is the von Kármán constant, and $z_0$ is the aerodynamic roughness length. The resulting fluxes are then transformed back into the global Cartesian coordinate system via a rotation tensor:

$$\tau_m = A^T \tau_{m,\text{local}} A, \tag{31}$$

where the local momentum flux tensor is defined as

$$\tau_{m,\text{local}} = \begin{bmatrix} 0 & 0 & \overline{u''w''}_{0,\text{local}} \\ 0 & 0 & \overline{v''w''}_{0,\text{local}} \\ \overline{u''w''}_{0,\text{local}} & \overline{v''w''}_{0,\text{local}} & 0 \end{bmatrix}, \tag{32}$$

and $A$ is the local-to-global transformation matrix given by

$$A = \begin{bmatrix} \cos\varphi_x & 0 & \sin\varphi_x \\ 0 & \cos\varphi_y & \sin\varphi_y \\ -\sin\varphi_x\cos\varphi_y & -\cos\varphi_x\sin\varphi_y & \cos\varphi_x\cos\varphi_y \end{bmatrix}. \tag{33}$$

## 4  Code framework and solving procedure

Figure 1 illustrates the overall code architecture and solution procedure of PALM-Sigma. The workflow begins by reading the input configuration parameters, defining the computational grid, and initializing all model variables. Once initialized, the

solver enters the main time integration loop to advance the flow field in time.

Within each time step, the prognostic solver first computes the tendencies of the velocity components by integrating the momentum equations, i.e., Eq. (10), (11), and (12). In parallel, it updates the wave field and calculates the associated sigma coordinate metrics. These updated metrics and wave boundary conditions are used to impose the effects of the moving wavy surface. The simulation then proceeds to the pressure solver stage, where the divergence of the predicted velocity field is

computed, the Poisson equation Eq. (24) is solved, and the velocity field is corrected to enforce the incompressibility constraint.

One iteration of the time-stepping loop is completed at this point. The solver continues looping through these steps until the prescribed simulation end time is reached. Finally, all simulation data are written to output files. For clarity, the main steps of the algorithm are summarized as follows:

1. Read input parameters and initialize the model.

2. Predict the intermediate velocity field using the prognostic equations.





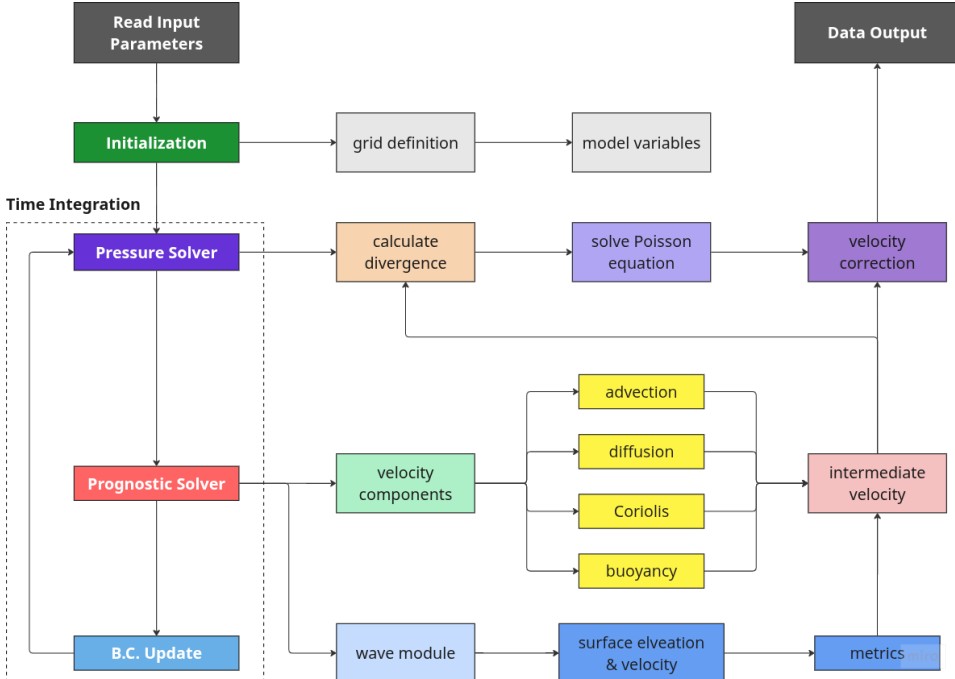

**Figure 1.** Schematic overview of the framework and workflow of the PALM-Sigma code.

3. Update the wave field and sigma coordinate metrics for boundary conditions and grid transformation.

4. Compute the velocity divergence and solve the pressure Poisson equation.

5. Correct the velocity field to satisfy the continuity constraint.

6. Repeat steps 2–5 for each timestep until the simulation end time.


7. Finalize and output simulation results.

## 5 Simulation setup

In this study, we designed four simulation cases under identical geostrophic wind conditions, with $U_g = 5\,\mathrm{m/s}$, but with varying wave scenarios. Specifically: a no-wave reference case (NW), a wind-following wave case (FW), a wind-opposing wave case (OW), and a steady (non-propagating) wave case (SW). In FW, OW, and SW, the wave parameters are identical: a wave height of $A = 1.6\,\mathrm{m}$ and a wavelength of $\lambda = 100\,\mathrm{m}$. Assuming linear deep-water wave theory, this corresponds to a wave period of $T = 8\,\mathrm{s}$ and a phase speed of $c = 12.5\,\mathrm{m/s}$. The only difference between FW and OW is the wave propagation direction relative to the wind. Neutral atmospheric stability is maintained by initializing a uniform potential temperature of 283 K from the surface up to 600 m, topped by a 100 m-thick inversion layer characterized by a lapse rate of 3.5 K/km.





The computational grid employs a horizontal resolution of $\Delta x = \Delta y = 3/64\lambda = 4.6875$ m. In the vertical direction, the
grid spacing is uniform at $2$ m from the surface up to $64$ m height. Above this level, the grid spacing gradually expands with
a constant stretching ratio of $1.025$, reaching a maximum cell size of $16$ m. The total grid consists of $256 \times 256 \times 128$ cells,
resulting in a domain size of $1200\,\text{m} \times 1200\,\text{m} \times 850\,\text{m}$. The simulation timestep is dynamically adjusted to satisfy the Courant-
Friedrichs-Lewy (CFL) condition, and it stabilizes at approximately $\Delta t = T/12$. Each simulation was integrated for a total of
$20$ h. The final hour of output was used for calculating time-averaged vertical profiles, while the last $30$ min (corresponding
to approximately $225$ wave periods) were used to compute and visualize wave-phase-averaged fields, including wave-induced
velocity components, pressure, and momentum fluxes.

## 6  Results

### 6.1  Flow field

Figure 2 shows the instantaneous fluctuations of the horizontal velocity at a height of $\lambda/5$, highlighting the near-surface flow
structures under different bottom boundary conditions. Figure 2(a) depicts the reference NW case, where elongated stream-
wise streaks characteristic of atmospheric boundary layer turbulence are evident. In contrast, Fig. 2(b)-(d) all exhibit distinct
and highly organized wavy flow structures that reflect the imposed sinusoidal wave geometry. Among these cases, the wind-
opposing wave (OW) scenario produces the most pronounced wave-induced fluctuations, which strongly modulate and dom-
inate over the background turbulence. The wind-following wave (WF) case also shows prominent wavy structures, although
their amplitude is slightly reduced compared to OW. The steady wave (SW) case presents weaker and less coherent wavy
patterns, where wave-induced fluctuations coexist with and are partially masked by turbulent motions.

Figure 3 presents snapshots of vertical velocity fluctuations across the $x$-$z$ plane, emphasizing the vertical extent of wave
impacts on the ABL flow. The field of $w$ component over a steady wavy surface behaves comparably to a rough fixed boundary,
with broadly similar turbulence characteristics as shown in Fig. 3(a) and (d), respectively. By contrast, the cases with moving
waves (panels b and c) display markedly different features. In the wind-following wave case, the near-surface flow exhibits
pronounced, wave-coherent vertical oscillations that remain organized along the wavy boundary. However, these structures
decay rapidly with height and largely vanish beyond about half a wavelength, leaving weaker turbulence aloft. In the wind-
opposing wave case, the wave-induced motions are strongly enhanced compared to all other cases. The fluctuations are not
only more intense near the surface but also penetrate much higher into the upper boundary layer under strong turbulent mixing
effects, substantially modifying the turbulent structure throughout the entire ABL depth.

As illustrated in Fig. 4, the presence of surface waves introduces pronounced oscillations in both the horizontal and vertical
velocity components along the wave surface across all three cases. In the left column, the phase-averaged horizontal velocity
$\langle u \rangle$ has distinct flow patterns depending on the wave condition. For the wind-following wave (FW) case, acceleration occurs
above wave troughs, with a peak velocity exceeding the geostrophic wind speed ($u_g$) at approximately $z \approx \lambda/5$. In contrast,
deceleration is observed above the wave crests. The wind-opposing wave (OW) case exhibits the opposite behavior: flow





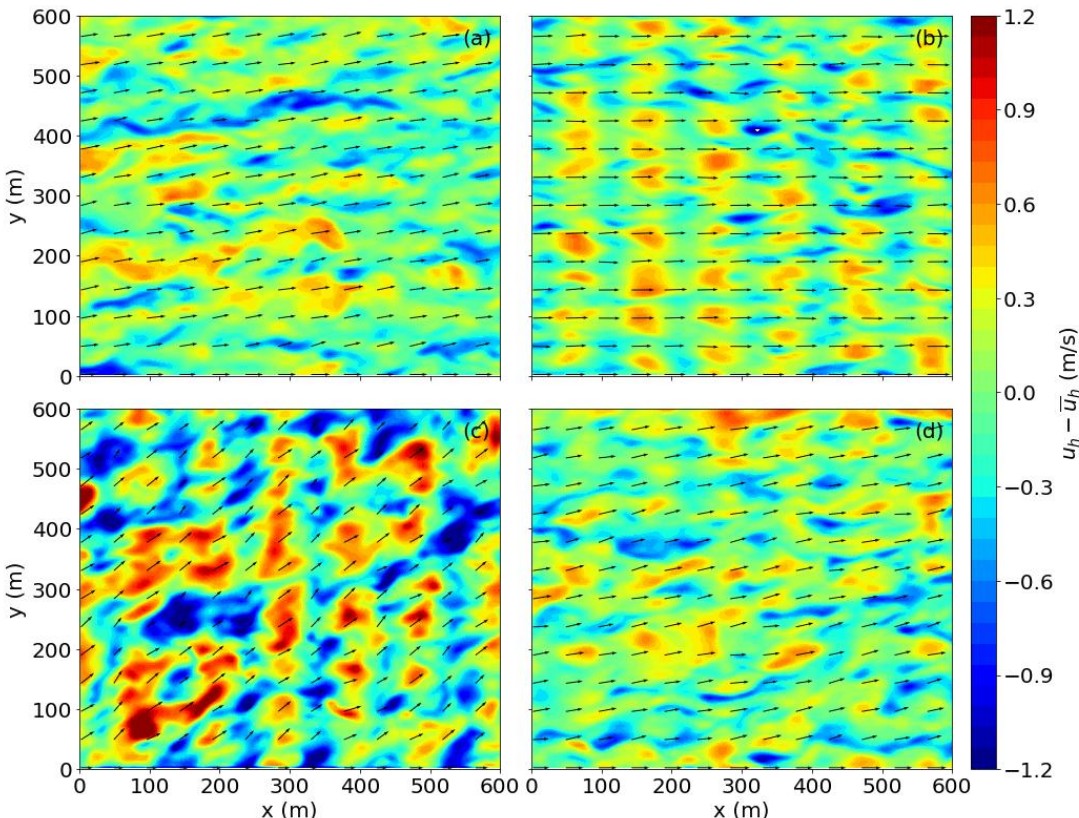

**Figure 2.** Instantaneous fluctuations of horizontal velocity in the $x$-$y$ plane at a height of $\lambda/5$ for four cases: (a) no wave (NW), (b) wind-following wave (WF), (c) wind-opposing wave (OW), (d) steady wave (SW). The black arrows represent the horizontal velocity vectors.

deceleration above troughs and acceleration over crests. A similar spatial pattern is observed in the steady wave (SW) case, though with considerably reduced intensity, indicating weaker influences on the flow by fixed surface bumps.

In the right column, the vertical velocity $\langle w \rangle$ further highlights the impact of wave motion. For the FW case, upward motion is found above the windward slopes, while downward motion appears above the leeward sides. This behavior reverses

in the OW and SW cases. The magnitude of $\langle w \rangle$ in the FW and OW cases is nearly an order of magnitude larger than in the SW case, underscoring the importance of wave propagation in enhancing vertical motion. The vertical profiles of $\langle w \rangle$ decay exponentially with height, becoming negligible above $z \approx \lambda$, consistent with theoretical and observational expectations for wave-induced motions (Makin et al., 1995; Högström et al., 2015; Wu et al., 2018). These simulated wave-induced flow structures in Fig. 4 show strong agreement with the LES results presented by Sullivan et al. (2008).



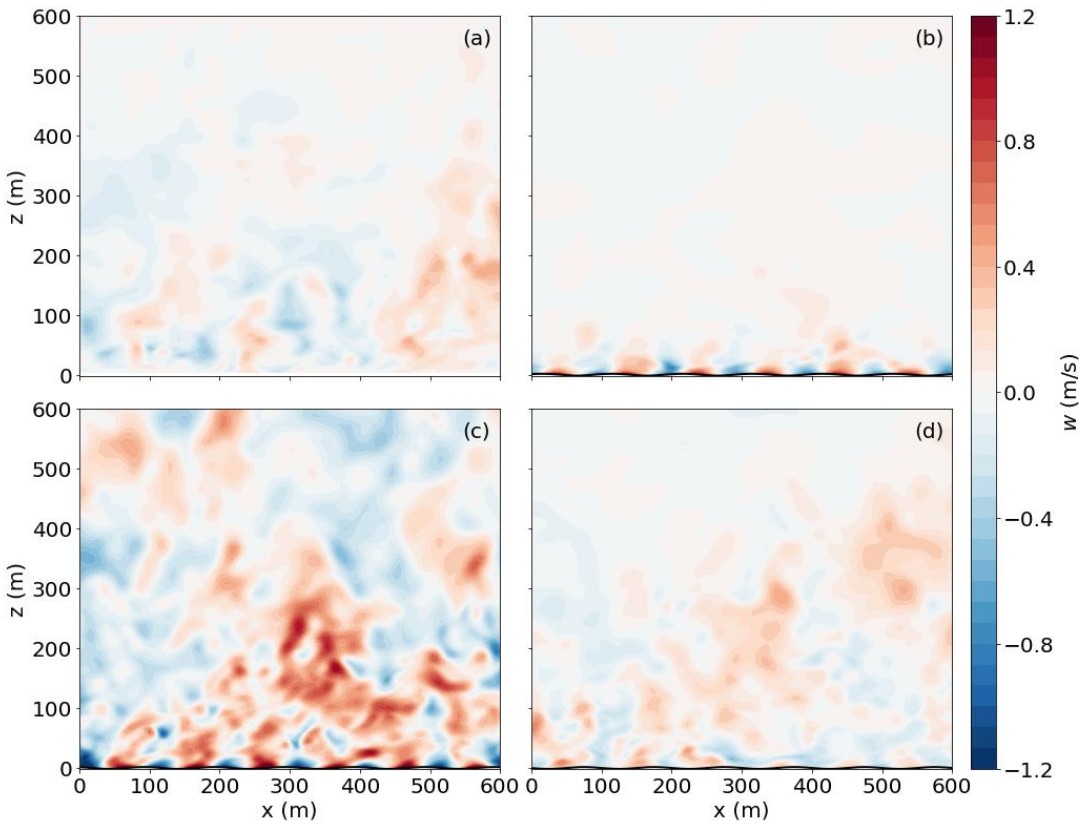

**Figure 3.** Instantaneous vertical velocity in the $x$-$z$ plane for four cases: (a) no wave (NW), (b) wind-following wave (WF), (c) wind-opposing wave (OW), (d) steady wave (SW).

Figure 5 presents the phase-averaged pressure fields in the $x$-$z$ plane, showing strong pressure gradients near the wave surface that diminish rapidly with height. Across all wave cases, regardless of wave direction or motion, the pressure exhibits a consistent spatial pattern: positive pressure zones are located in the wave troughs, while negative pressure zones occur above the wave crests. However, the magnitude of pressure variation varies significantly among cases. The OW case exhibits the strongest pressure fluctuations, followed by the FW case, where the peak amplitude is roughly half that of OW. The SW case shows much weaker pressure variation, approximately an order of magnitude smaller. The pressure profiles along the wave surface (bottom right panel) further reveal differences in the phase relationship between pressure and surface elevation. In the SW case, pressure fluctuations are nearly out-of-phase with the wave elevation, implying a symmetric pressure distribution around the wave crest and trough. Nevertheless, in the FW case, the pressure maximum is shifted upwind (toward the front





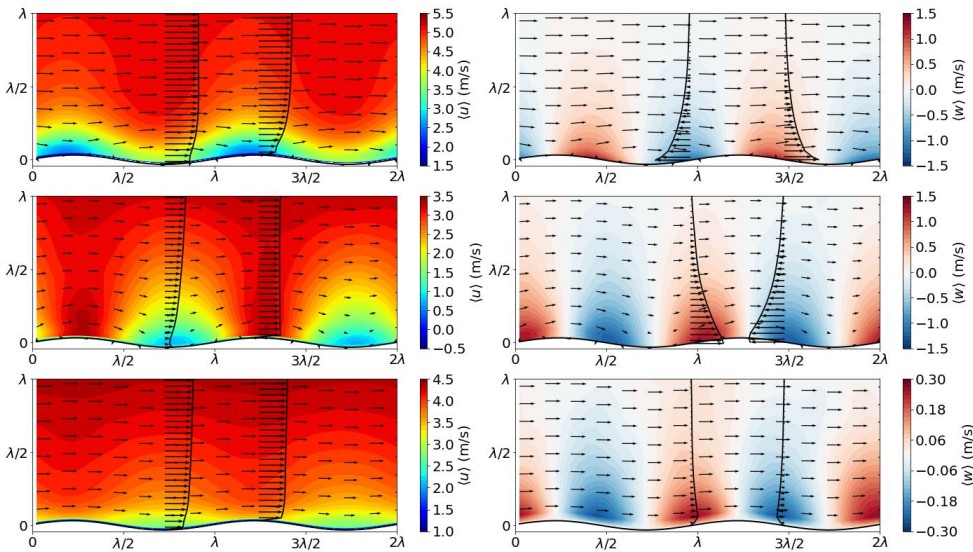

**Figure 4.** Phase-averaged velocity fields in the $x$–$z$ plane. Left column: Horizontal velocity component $\langle u \rangle$, with vertical profiles plotted above the wave troughs and crests. Right column: Vertical velocity component $\langle w \rangle$, with vertical profiles above the windward and leeward wave slopes. Black arrows in the flow field represent the phase-averaged velocity vectors.

face of the wave), occurring slightly upstream of the trough. Conversely, in the OW case, the pressure peak is displaced towards
downstream of the trough. This asymmetry in pressure distribution correlates with the patterns of vertical velocity discussed in Fig. 4, where the upward motion and high pressure are concentrated on the front slope of the wave (the side facing wave propagation). Such asymmetric pressure patterns lead to the so-called form stress, which plays a critical role in modifying the vertical profiles of both mean wind speed and direction above the wave surface.

## 6.2    Spectral characteristics

This section examines the spectral characteristics of the marine atmospheric boundary layer flow under the influence of different wave conditions. The spectral analysis is based on velocity data collected from a $33 \times 33$ probe matrix with uniform horizontal spacing of 9.375 m, sampled at a frequency of 2 Hz over a duration of half an hour. Measurements were taken at multiple vertical levels to assess the height-dependent response of the flow.

     Figure 6 presents the power spectral density (PSD) curves of the three velocity components at three representative heights
($H = \lambda/5, \lambda/2,$ and $\lambda$) for all wave cases. The impact of surface waves on the MABL flow manifests in two primary ways. First, in the $u$ and $w$ spectra, distinct peaks appear at the fundamental wave frequency and its first two harmonics below $H = \lambda/2$. These spectral peaks indicate the presence of coherent wave-induced structures aligned with the direction of wave propagation.





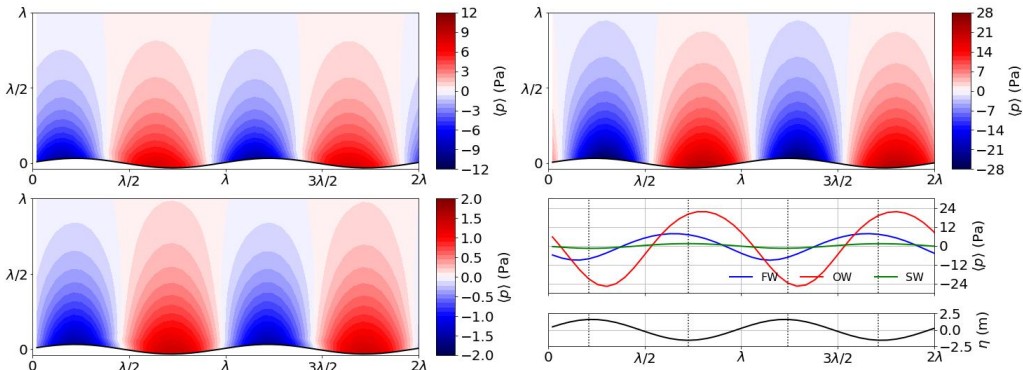

**Figure 5.** Phase-averaged pressure fields $\langle p \rangle$ in the $x$–$z$ plane for the three wave scenarios: wind-following wave (FW, top left), wind-opposing wave (OW, top right), and steady wave (SW, bottom left). The bottom-right panel presents the corresponding pressure distributions along the wave surface ($z = \eta$) for all cases, plotted together for direct comparison, alongside the wave surface elevation $\eta(x)$ in black. Dotted lines indicate the trough and crest positions.

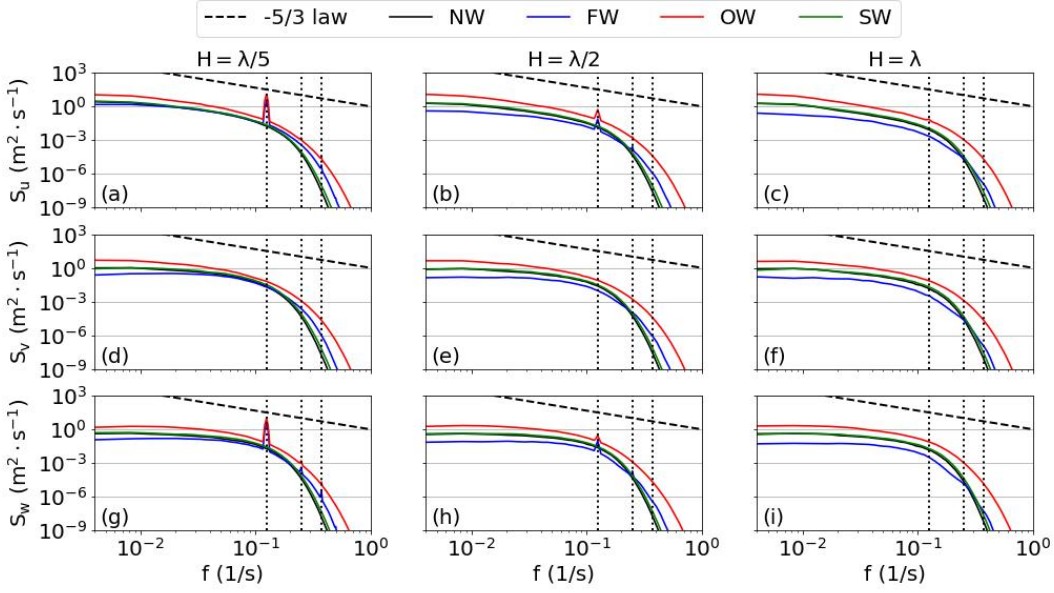

**Figure 6.** Horizontally averaged power spectral density curves of the three velocity components-$u$ (top row), $v$ (middle row), and $w$ (bottom row), at three vertical heights: 20 m (left), 40 m (middle), and 100 m (right), under four different wave conditions (NW: no wave, FW: following wave, OW: opposing wave, SW: steady wave). The black dashed line indicates the $-5/3$ slope of the inertial subrange. Vertical dotted lines denote the wave frequency and its first two harmonics.




No such peaks are observed in the $v$ component, suggesting that the wave effects are anisotropic and predominantly oriented in the streamwise and vertical directions. At $H = \lambda$, approximately one wavelength above the surface, these wave-induced
signatures vanish, indicating limited vertical penetration of wave-coherent motions.

Second, surface waves substantially modify the overall turbulence energy distribution. Across all three velocity components and frequency bands, the opposing wave case exhibits significantly elevated PSD levels, indicating enhanced turbulence production throughout the boundary layer. In contrast, the FW case consistently exhibits reduced spectral energy, implying suppression of turbulence. These trends are not limited to the wave boundary layer (WBL) but extend across the entire vertical
domain, consistent with the vertical profiles of turbulent kinetic energy (TKE) shown in Fig. 8(f). Notably, the steady wave case yields PSD levels nearly identical to those observed over a flat surface, reinforcing the critical role of wave motion, rather than wave geometry alone, in shaping the boundary-layer turbulence dynamics.

The power spectral density analysis indicates the presence of wave-induced temporal oscillations in the streamwise and vertical velocity components, especially below half a wavelength in height. To further investigate spatial coherent structures
arising from wave motions, Fig. 7 presents vertical coherence and phase spectra between velocity components separated by 20 m. The first row displays the $u$-$u$ coherence spectra. In the NW and SW cases, coherence decays exponentially with increasing frequency and vanishes beyond $f \approx 0.05$ Hz. In contrast, the FW and OW cases exhibit distinct coherence peaks approaching unity at the wave frequency when $H = \lambda/5$. Peaks at the second harmonic are also observed, but with a weaker intensity. Interestingly, wave-induced coherence remains strong (up to 0.5) even at $H = \lambda/2$, indicating that wave-induced coherent
motions persist well above the surface layer. The corresponding phase spectra (second row) reveal minimal phase difference between $u$ components at different vertical levels at wave-related frequencies. The standard deviation (dotted lines) increases sharply beyond $f \approx 0.05$ Hz, indicating loss of correlation and resulting in random phase differences. However, around the wave frequencies and harmonics, the phase variance collapses into an "hourglass" shape, further confirming the in-phase, vertically coherent structures induced by wave motion.

The third row shows $u$-$w$ coherence, generally weaker in the low-frequency range than $u$-$u$, but exhibiting equally sharp peaks at wave frequencies in the FW and OW cases at $H = \lambda/5$. Notably, at $H = \lambda/2$, the FW case shows stronger wave-related coherence than the OW case. This implies that wave-induced $u$–$w$ coupling becomes more prominent when the background turbulence is weaker. The bottom row presents $u$-$w$ phase spectra. Unlike the in-phase behavior of $u$-$u$, the $u$-$w$ phase shift exceeds $\pi/2$ at low frequencies across all cases, suggesting that their product ($u'w'$) tends to fall into quadrant 2 and quadrant
4, corresponding to ejection and sweep events. This contributes to a negative vertical momentum flux. As frequency increases, the phase difference linearly decreases to zero around $f \approx 0.2$ Hz. Notably, the FW and OW cases show sharp $\pm\pi/2$ phase shifts precisely at the wave frequency and harmonics, implying that wave-induced $u$-$w$ structures are primarily out-of-phase, and thus do not significantly contribute to vertical turbulent momentum transport.

### 6.3 Vertical structure

We have previously discussed the characteristics of wave-induced velocity and pressure oscillations. These oscillations exert significant impacts on the vertical structure of marine atmospheric boundary layer flows.



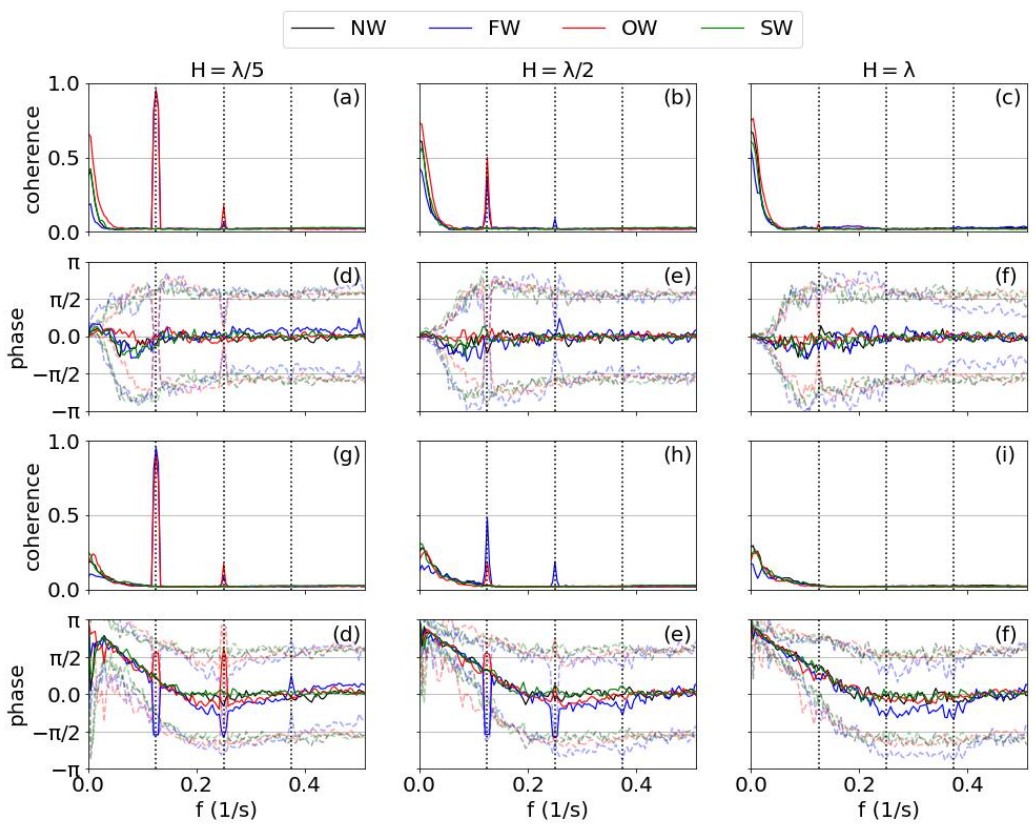

**Figure 7.** Vertical coherence and phase spectra for velocity components at a vertical separation of 20 m. The top two rows show the coherence and phase of the streamwise velocity ($u$–$u$), while the bottom two rows show the coherence and phase between streamwise and vertical velocity components ($u$–$w$). Each column corresponds to a different vertical location of the lower probe: $H = \lambda/5$ (left), $H = \lambda/2$ (middle), and $H = \lambda$ (right). Black, blue, red, and green lines represent the NW, FW, OW, and SW cases, respectively. Dashed vertical lines indicate the wave fundamental frequency and its harmonics. The dashed lines in the phase spectra mark the $\pm 1$ standard deviation.

First, surface waves contribute directly to the vertical momentum flux through the asymmetric pressure distribution along the wavy surface, as demonstrated in Fig. 5(d). Specifically, the integral average of the horizontal pressure projection, expressed as

$$M_w = \frac{1}{\lambda} \int\limits_0^\lambda p\zeta_x/\zeta_z \, dx, \tag{34}$$

acts effectively as a vertical momentum flux and can become the dominant dynamical forcing in the wave boundary layer. The velocity field is also strongly modulated by wave motions, but the streamwise and vertical wave-induced velocity components are predominantly out of phase, as shown in the last row of Fig. 7. As a result, their contribution to the momentum flux is





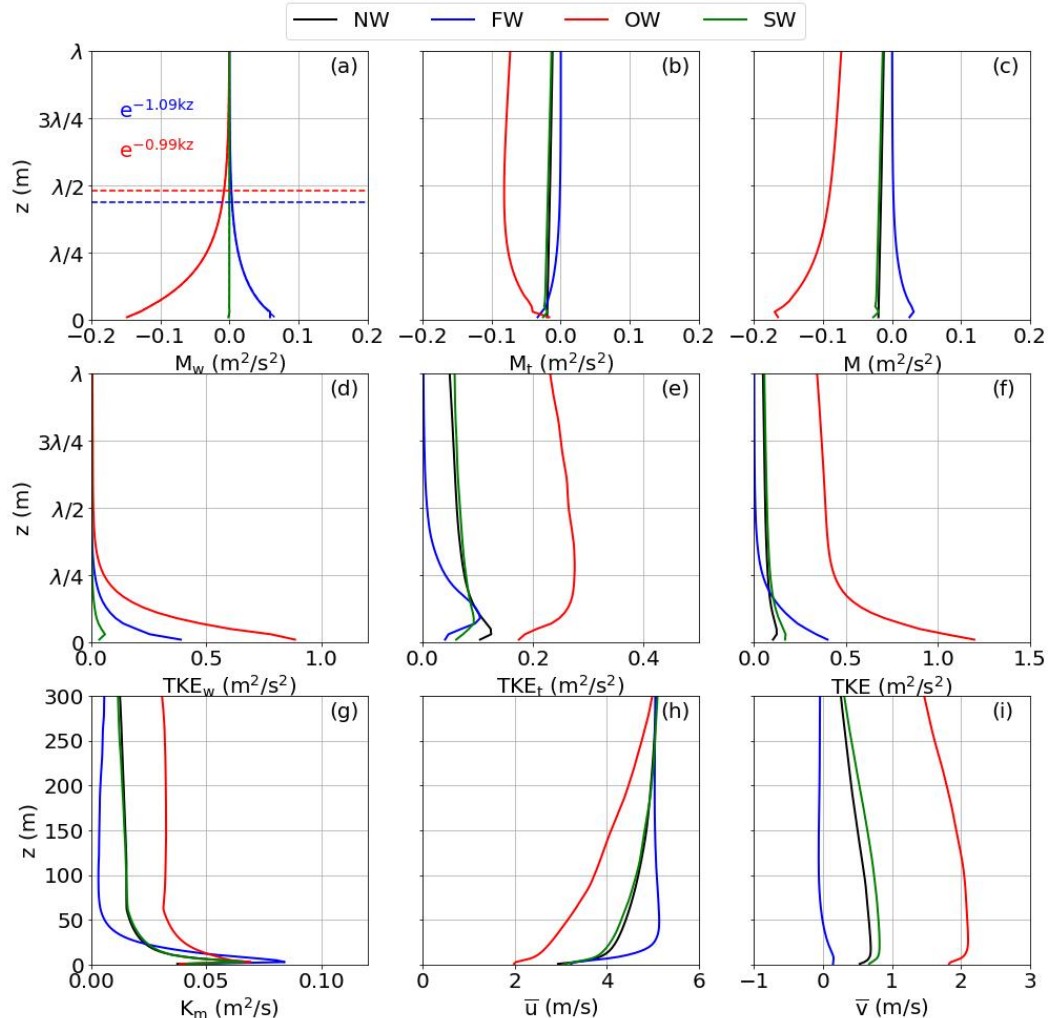

**Figure 8.** Vertical profiles of flow and turbulence statistics under different wave conditions. Top row: (a) Streamwise wave-induced momentum flux; (b) Turbulent momentum flux; (c) Total streamwise vertical momentum flux. Middle row: (d) wave-induced kinetic energy; (e) turbulent kinetic energy excluding wave contributions; (f) Turbulent kinetic energy. Bottom row: (g) Eddy diffusivity; (h) Mean streamwise velocity $\overline{u}$; (i) Mean crosswise velocity $\overline{v}$. Black, blue, red, and green curves represent the NW, FW, OW, and SW cases, respectively. In subplot (a), fitted exponential decay functions are also shown, along with markers indicating the heights where the wave-induced momentum flux decreases to 95% of its surface value.



negligible. Therefore, the total vertical momentum flux, here denoted as $M$, can be decomposed into two parts: the wave-induced momentum flux $M_w$ and the turbulent momentum flux $M_t$. The first row of Fig. 8 presents their vertical profiles. In subplot (a), it is evident that $M_w(z)$ decays exponentially with height in both the FW and OW cases. Remarkably, this decay is well described by the exponential function $f(z) = e^{-akz}$. The fitted decay coefficients are $a = 1.09$ for the FW case and $a = 0.99$ for the OW case, respectively. The fitted curves are plotted as dashed lines, but they are almost indistinguishable from the simulation results due to the excellent agreement. This provides strong support for the assumption of an exponential decay with $a = 1.0$, adopted in our earlier parameterization study (Ning and Bakhoday-Paskyabi, 2025). The dashed lines indicate the estimated heights of the wave boundary layer in the two cases, defined as the levels where $M_w$ decreases to $5\%$ of its surface value. These heights are both close to half the wavelength, with the WBL in the OW case extending slightly higher than in the FW case.

The turbulent momentum flux is also indirectly modulated by the presence of waves, as shown in subplot (b). Compared to the case with a flat surface (NW), $M_t$ in the FW case is larger near the surface but decays rapidly to nearly zero by a height of $\lambda/4$. This reduction is related to the much weaker wind shear over the wave boundary layer in the presence of following waves. In contrast, the OW case exhibits an enhancement of downward momentum flux throughout the boundary layer due to the additional surface drag imposed by the opposing wave. Figure 8(c) further illustrates that the influence of waves on the total vertical momentum flux extends well above the top of WBL. Additionally, the SW case, with a fixed wave surface, does not induce significant changes in the momentum flux profiles relative to the NW case.

Similarly, the total turbulent kinetic energy ($E$) can be decomposed into the wave-induced kinetic energy ($E_w$) and the non-wave-related turbulent kinetic energy ($E_t$). Their vertical profiles are shown in the second row of Fig. 8 to illustrate the influence of waves on TKE. $E_w$, associated with wave-coherent flow motions, also exhibits an exponential decay with height. However, the decay rate is higher than that observed for the momentum flux, and its magnitude is highly sensitive to the relative orientation between the wind and wave directions rather than to the wave characteristics alone. Under the same wave amplitude and phase speed, $E_w$ at the first grid level in the wind-opposing wave case (accounts for $73.8\%$ of $E$), is nearly twice as large as in the wind-following wave case, where $E_w$ represents $97\%$ of $E$ due to the much lower total TKE. By comparison, the case with a steady wavy surface shows a smaller contribution, with $E_w$ comprising only about $20\%$ of $E$. The modification of TKE by waves further affects the parameterization of eddy diffusivity, as shown in Fig. 8(g). The larger values of $K_m$ in the OW case indicate stronger turbulent mixing of scalars and momentum, while in contrast, $K_m$ in the FW case is nearly an order of magnitude smaller, reflecting a more stratified boundary layer structure. These trends are also evident in Fig. 3(b).

Subplots (h) and (i) present the vertical profiles of the streamwise and crosswise velocity components, respectively. In the FW case, the streamwise velocity profile exhibits a pronounced knee-shaped turn at $z \approx \lambda/4$, above which the wind becomes nearly uniform. This characteristic profile is commonly observed when low wind speeds are aligned with fast-propagating underlying waves (Högström et al., 2013). Conversely, in the OW case, the streamwise wind speed is substantially reduced from the surface up to 300 m, primarily due to surface drag increasing to nearly eight times that observed in the NW and SW cases. Although wave-coherent motions are confined to the $x$-$z$ plane, the profile of $v$ is strongly affected by waves. This occurs





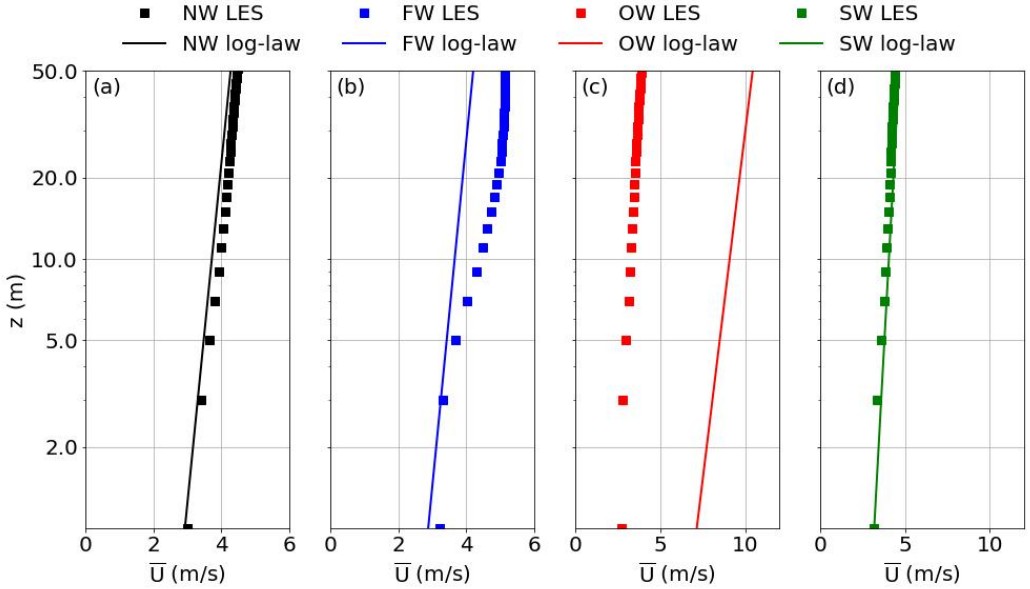

**Figure 9.** Comparison between the simulated mean wind speed profiles (by square markers) and the corresponding logarithmic wind profiles calculated by MOST (by solid lines) for each case: (a) NW, (b) FW, (c) OW, and (d) SW. The friction velocity used for MOST is derived from a 10 m height.

because changes in $u$ alter the Coriolis force and reshape the Ekman spiral, thus resulting in a clockwise wind veer in the FW case and an anticlockwise rotation in the OW and SW cases relative to the case without waves.

### 6.4 Surface layer under wave impacts

The Monin-Obukhov Similarity Theory (MOST) has been widely employed to estimate the wind profile and stress within the surface layer and has shown good agreement with observations under a variety of conditions. Under neutral stratification, the wind profile predicted by MOST follows a logarithmic form:

$$\overline{U}(z) = \frac{u_*}{\kappa} \ln \left( \frac{z}{z_0} \right), \tag{35}$$

where the friction velocity $u_* = \sqrt{|M|}$ is assumed constant. However, when wind-wave coupling is not in equilibrium, particularly in the presence of fast-moving swell beneath relatively low winds, as in our FW and OW cases, the Eq. (35) may no longer accurately represent the actual wind profile due to additional wave-induced flow contributions. Figure 9 compares the MOST-predicted logarithmic profiles to our simulation results, demonstrating the extent to which wave-dominated wind deviates from classical similarity theory.





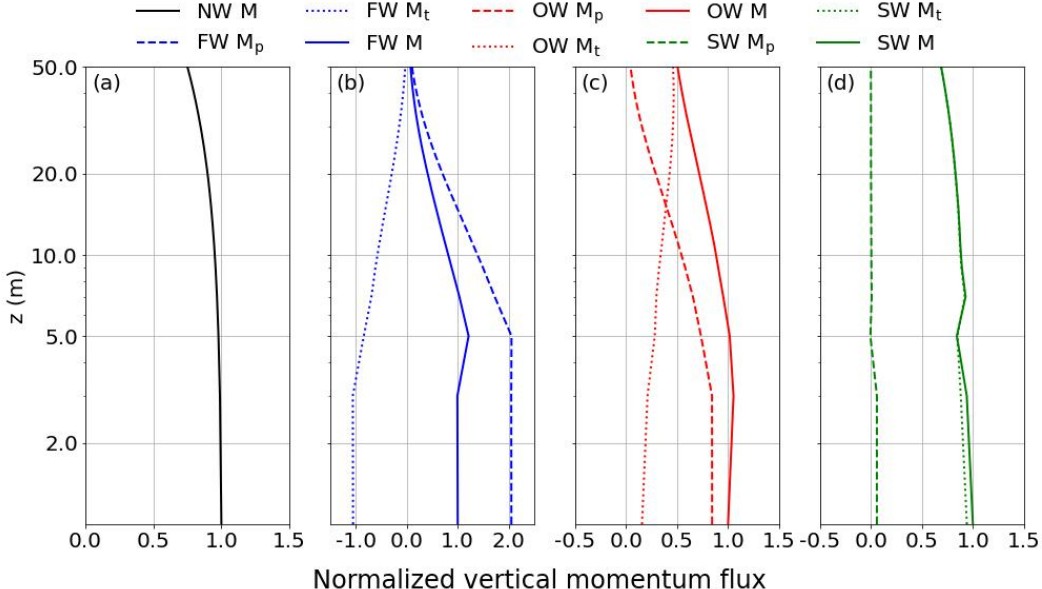

**Figure 10.** Vertical profiles of normalized momentum flux components in the four cases: (a) NW, (b) FW, (c) OW, and (d) SW. Solid lines denote the total vertical momentum flux, dashed lines show the wave-induced contribution, and dotted lines indicate the turbulent contribution. All fluxes are normalized by the surface momentum flux in each case.

Here, the friction velocity is computed based on the momentum flux along the mean wind direction at the reference height $z = 10$ m, i.e.

$$u_{*,10} = \sqrt{\frac{|\boldsymbol{M}_{10} \cdot \overline{\boldsymbol{U}}_{10}|}{|\overline{\boldsymbol{U}}_{10}|}}. \tag{36}$$

As expected, the wind profile in the NW case forms an approximately straight line on the logarithmic scale, closely following the log-law predicted by MOST. In contrast, substantial deviations appear in both the wind-following and wind-opposing wave cases. In the FW case, the wind speed begins to exceed the logarithmic profile just above 3 m, with the wave-driven component contributing 19.6% more wind speed than the log-law prediction at 10 m height, reaching a maximum excess of 26.8% at 23 m. Conversely, in the OW case, the log-law significantly over-predicts the near-surface wind: at the wave surface, it estimates wind speeds 1.6 times higher than those resolved in the simulation. The discrepancy further increases with height. In contrast, the SW case shows excellent agreement with the log-law until a height of 50 m, indicating that a steady wave field effectively behaves as a roughness element that can be effectively parameterized by an equivalent roughness length.

An important premise of MOST is the assumption of a constant momentum flux within the surface layer. To evaluate the validity of this assumption under different wave conditions, Fig. 10 presents the vertical profiles of the partitioned momentum flux. Overall, the constant flux approximation holds reasonably well up to a height of 10 m in all cases, particularly for the case without waves. At 10 m, the total momentum flux remains at 81.7%, 88.3%, and 88.4% of its surface value in the FW, OW,



**Table 2.** Surface layer parameters obtained from LES results.

| Case ID | $U_{10}$ (m/s) | $M_{w,10} \times 10^2$ (m²/s²) | $M_{t,10} \times 10^2$ (m²/s²) | $M_{10} \times 10^2$ (m²/s²) | $u_{*,10}$ (m/s) | $c/u_{*,10}$ | $C_D \times 10^3$ | $\beta \times 10^4$ |
|---|---|---|---|---|---|---|---|---|
| NW | 3.96 | | -1.89 | -1.89 | 0.137 | | 1.21 | |
| FW | 4.38 | 3.47 | -1.64 | 1.83 | 0.135 | 92.6 | -0.96 | -0.59 |
| OW | 3.27 | -6.62 | -4.68 | 11.30 | 0.336 | 37.2 | 10.58 | -1.48 |
| SW | 3.90 | -0.01 | -2.24 | -2.25 | 0.150 | | 1.48 | |

and SW cases, respectively. These results are consistent with the observations reported by Högström et al. (2013), and support the parameterization approaches proposed by Semedo et al. (2009); Wu et al. (2018); Ning and Paskyabi (2024), where the wave-induced stress is incorporated into the log-wind formulation. However, it is evident that in both FW and OW cases, the momentum flux within the near-surface layer is strongly dominated by the wave-induced component. This dominance causes the total flux to decay substantially above 10 m, as the contribution from wave-induced stresses diminishes rapidly over this

height.

As the wave propagates beneath the wind, the energy exchange between wind and wave is done by the form stress generated from the pressure distribution along the wind-wave interface. The associated wave damping rate can be expressed as the product of this stress and the wave phase speed:

$$\beta = -\frac{\boldsymbol{\tau}_w}{\rho_w} \cdot \frac{\boldsymbol{c}}{\frac{1}{2}gA^2} = -\frac{\frac{\rho_a}{\lambda}\int_0^\lambda p\zeta_x/\zeta_z dx}{\frac{1}{2}\rho_w\omega A^2} \tag{37}$$

Table 2 summarizes the wave damping rates computed for cases FW and OW, together with other key parameters characterizing the surface layer derived from the LES results.

The wave age $c/u_{*,10}$ in both FW and OW cases greatly exceeds the threshold value of 20 (Cohen and Belcher, 1999), implying that they are clear wave-driven wind scenarios. Notably, the computed wave damping rates are on the order of $10^{-4}$, in good agreement with the fitted values reported by (Semedo et al., 2009). Furthermore, despite identical background

geostrophic wind and wave parameters, the damping rate $\beta$ in case OW is approximately 2.5 times higher than in case FW, underscoring the strong dependence of wind-wave interaction on the relative alignment between wind and wave propagation. The wind-opposing wave configuration produces substantially stronger momentum and energy exchange across the interface, manifesting as enhanced surface drag, increased turbulence intensity, and a pronounced reduction in near-surface wind speeds. As a result, the drag coefficient $C_D$ increases markedly from $1.21 \times 10^{-3}$ in the NW case to $1.06 \times 10^{-2}$ in OW, while in FW

it is reduced to $-9.6 \times 10^{-4}$, where the negative sign denotes an upward momentum flux from the waves to the atmosphere. These trends are qualitatively consistent with the LES findings reported by (Jiang et al., 2016).



## 7 Conclusions

In this work, we developed a wave-phase-resolved large-eddy simulation solver employing a sigma coordinate system, implemented within the framework of the PALM model system, to investigate wind-wave interactions and their impacts on marine atmospheric boundary layer flows. The newly implemented surface-following coordinate formulation enables direct resolution of the flow dynamics induced by the time-varying wavy surface geometry. To evaluate the model's performance, we configured four simulation cases, including a flat surface, wind-following waves, wind-opposing waves, and steady waves. Clear peaks observed in the power and coherence spectra demonstrate the presence of strong wave-coherent flow structures in the wave boundary layer. Moving waves also modify the vertical structure of the boundary layer by imposing momentum fluxes via the asymmetric pressure distributions along the wave surface. The magnitude and orientation of these fluxes are strongly dependent on the combined wind and wave conditions. Although the wave-induced velocity and pressure fluctuations decay exponentially with height and become negligible above approximately half a wavelength, their influence on the wind speed and turbulent kinetic energy profiles extends much farther into the boundary layer.

The solver successfully reproduces typical wave-driven flow features observed under low-wind with strong swell scenarios, and highlights the essential role of wave motion and direction in shaping boundary-layer dynamics. Further validation work is planned, including coupling the solver with wave models to represent spectral wave fields and comparing results against observational data. Additionally, investigating wave impacts under a broader range of wave age, wind-wave misalignment, and atmospheric stability conditions will be an important focus of future studies. Overall, this model offers a high-fidelity tool for atmosphere-wave coupled simulations, facilitating both fundamental investigations of wind-wave interactions and the evaluation of parameterization approaches.

*Code and data availability.* The Simulation code used in this study, named PALM-Sigma v1.0, is developed based on the open-source large-eddy simulation (LES) model PALM v21.10 (Maronga et al., 2020), publicly available at https://gitlab.palm-model.org/releases/palm_model_system (last access: 11 Sep 2025). The implementation of the sigma coordinate system and related modified PALM modules for wave-phase-resolved modeling is enveloped in the USER_CODE folder, which is a dedicated interface for user-specific extensions and can be directly compiled and executed within the standard PALM framework. A stable, citable release of PALM-Sigma v1.0, including the related source files and a user's guide, is archived at Zenodo https://doi.org/10.5281/zenodo.17246634 (Ning, 2025a). The most up-to-date development version of PALM-Sigma is maintained on GitHub at https://github.com/XuNing-GBY/PALM-Sigma (last access: 11 Sep 2025). The integration of PALM-Sigma into the standard PALM distribution is planned in close collaboration with the PALM developer team at Leibniz University Hannover for a future release. All simulation input files, configurations, and output data used in this study are archived and openly available at https://doi.org/10.5281/zenodo.17101459 (Ning, 2025b).



*Author contributions.* XN: Conceptualization, Code development, Methodology, Investigation, Formal analysis, Writing – original draft, Writing – review & editing, Validation. MBP: Conceptualization, Methodology, Final review, Planning and supervision of the work, Project administration, Funding acquisition.

*Competing interests.* The author has declared that there are no competing interests.

*Acknowledgements.* This work is part of the LES-WIND project, and the authors gratefully acknowledge funding from the academic agreement between the University of Bergen and Equinor. The Large Eddy Simulations were conducted using the High-Performance Computing facilities of the Norwegian e-infrastructure UNINETT Sigma2 (project number NS9871K). The NIRD cloud service was utilized for data storage and processing. Additionally, we acknowledge the use of AI technology solely for enhancing text readability, including refining sentence structure and word choice, to ensure clarity and better comprehension for readers.



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
