# Peer review of "Implementation of a sigma coordinate system in PALM-Sigma v1.0 (based on PALM v21.10) for LES study of the marine atmospheric boundary layer"

_EGUsphere, 2025_

## Referee Comment (RC3)

Wave-atmosphere interactions are crucial for accurate weather and climate prediction. In this study, the authors developed a sigma coordinate system within the PALM model that directly resolves wave phases. Several case studies are employed to demonstrate the model's results. This modeling tool holds significant potential to provide deeper insights into wave-atmosphere interactions and to benefit the broader research community. Overall, the paper's structure is clear and easy to follow. However, the verification of the model results needs to be strengthened before I can recommend publication. Detailed comments are provided below:

- Section 5: Do you include the roughness length used to describe the drag from unresolved small-scale waves riding on the larger-scale resolved swell, as done in Peter Sullivan's model? If so, what is its value?
- Figure 3: When comparing your results for the OW case with those from Sullivan's OW case (Fig. 6 in their study), your results exhibit much larger wind fluctuations that extend to a much higher layer. Why is this?
- Wind profiles comparison: Your wind profiles for the OW and SW cases show substantial differences. However, in the results from Sullivan's code, these two experiments exhibit only small differences in the wind profile (See their Fig. 10). I suggest that the authors compare their model results with previous LES/DNS simulations to verify their findings.
- Energy budget: Did you check the energy budget of the LES simulations? Are they closed?

Minor comments:
- Page 9: L188: A is not defined
- Page 11, L240: A is wave amplitude, not wave height, right?
- L340: marine atmospheric boundary layer has been defined as MABL. Use it
- In Fig. 8, TKE_w should not be used for the wave kinetic energy, which is misleading. It should be WKE.
- Figure 4: I suggest changing the colormap in the left panel. As it stands, it is not easy to discern at which height the wind approaches the geostrophic wind described in the text.
- Y-axis clarification: Is the y-axis used in your figures based on height above the mean sea surface level or above the wave surface?

[Figure]

FIG. 10. Vertical profiles of the horizontal components of the mean wind $\langle \bar{u} \rangle$, $\langle \bar{v} \rangle$ for flow over waves. The spatial averaging is carried out along constant $\zeta$ surfaces, where $\zeta$ is the mean height above the wave. The nominal boundary layer depth $z_i = 400$ m. The cases are as follows: dotted line, no waves; squares, stationary bumps; diamonds, wind following waves; triangles, wind opposing waves; and dashed line, slight convection with wind following waves.

Sullivan, P.P., Edson, J.B., Hristov, T. and McWilliams, J.C., 2008. Large-eddy simulations and observations of atmospheric marine boundary layers above nonequilibrium surface waves. *Journal of the Atmospheric Sciences*, 65(4), pp.1225-1245.

---

## Author Comment (AC1)

**Response to Reviewer 2**

**Title:** Implementation of a sigma coordinate system in PALM-Sigma v1.0 (based on PALM v21.10) for LES study of the marine atmospheric boundary layer

**Manuscript number:** egusphere-2025-4390

We take this opportunity to thank the editor and reviewers of our paper for their kind collaboration to the improvement of this manuscript. We have taken into account all the concerns raised and we have made suggested modifications, marked by yellow background in the revised manuscript.

**Comments and responses**

**Reviewer #2**

**General comments:** The authors present the implementation of a sigma coordinate system in the LES code PALM and show its application to the marine atmospheric boundary layer. The authors show how their code development significantly improves the representation of the interaction between a wavy water surface and the marine boundary layer. Their comparison between different cases with and without waves with and without movement is done with great detail and reveals significant changes in the mean properties and the turbulent structure of the marine boundary layer if the new sigma coordinate system is used with moving waves. Their findings also agree with other results reported in the literature. Therefore, I recommend accepting the manuscript for publication in EGUsphere. However, I would like to ask the authors to consider the following comments.

**Response:** We greatly appreciate your recognition of our work and the constructive suggestions. We have carefully addressed each of your concern and have provided detailed responses below.

**Major Comments:**

1. p.3, l.63: "PALM" should be used as a fixed name, not as an abbreviation as stated in Maronga et al. (2020).

**Response:** We thank the reviewer for pointing this out. The expansion of PALM as "Parallelized Large-Eddy Model" has been removed, and PALM is now consistently treated as a fixed model name throughout the manuscript.

2. p.3, l.87: theta_v stands for the *virtual* potential temperature (in Table 1, it is already correctly defined).

**Response:** We thank the reviewer for pointing out this inconsistency. The definition of theta_v has been corrected to "virtual potential temperature" in the text, consistent with Table 1.

3. p.10, l.222: "In parallel, it updates the wave field [...]" I doubt that this happens in parallel (meaning that a part of the computing units does the update of the wave fields while other units integrate the prognostic equations) but more likely one after the other.

**Response:** We thank the reviewer for this clarification. We would like to express here that the velocity field and wave surface are updated independently without exchanging information. The term "in parallel" was indeed not precise and misleading in this context. The text has been revised to clarify that the velocity tendencies are computed first, followed by an update of the wave field and sigma coordinate metrics within each time step.

4. p.10, l.227: "Finally, all simulation data ware written to output files." In standard PALM, this is done within the time-stepping loop to allow, e.g., hourly data output.

**Response:** We thank the reviewer for this clarification. The text has been revised to reflect the standard PALM output strategy, where simulation data are written at user-defined output intervals during the time-stepping loop, and subsequently organized and aggregated into the final output files after completion of the simulation.

5. Fig.1: The figure does not show what is written in the text. On the left, the pressure solver and the prognostic solver should be switched. The Boundary-condition update is not mentioned in the text. Also, the flow structure is not that well represented. I recommend updating the figure to better show the program structure. An example would be Fig. 10 in Maronga et al. (2015, doi:10.5194/gmd-8-2515-2015) which shows the flowchart of an older version of PALM.

**Response:** We thank the reviewer for this detailed and helpful comment. Figure 1 and

the corresponding manuscript text have been revised to better reflect the program structure and time-stepping procedure of PALM-Sigma.

In the original version, the time-stepping loop was shown to start with the pressure solver because, in PALM, the pressure solver can be invoked immediately after initialization to improve the zero-divergence condition before time integration. However, following the reviewer's suggestion, we have revised Fig. 1 to place the prognostic solver ahead of the pressure solver, which better aligns with the standard fractional-step method used during time integration. To avoid ambiguity, we have added a clarifying note in the text (in parentheses) explaining this design choice.

Furthermore, the arrows within the prognostic solver have been modified to more accurately represent the numerical procedure. In particular, advection, diffusion, and the remaining tendency terms are now shown as being applied sequentially rather than in parallel, consistent with their actual execution order in the model.

Finally, the boundary condition update, which was previously only shown in the figure, is now explicitly described in the revised manuscript text and clearly indicated within the time-stepping loop in Fig. 1.

Overall, Fig. 1 has been redesigned to more clearly illustrate the solver sequence and program flow, following the general style of the PALM flowchart presented in Maronga et al. (2015), and the text has been revised accordingly to ensure full consistency between the figure and the description.

6. p.13, l.279: From my understanding of the figures, the words "windward" and "leeward" should be swapped in this sentence.

**Response:** We thank the reviewer for pointing this out. The side of the wave facing into the wind is "windward" and the side of the wave sheltered from the wind is "leeward." They are misused in the original manuscript. This has been corrected in the revised manuscript to ensure consistency with the flow patterns shown in the figures.

7. Fig.4: Please add which cases are represented in each row.

**Response:** We thank the reviewer for this suggestion. Panel labels (a) – (f) have been added to Fig. 4, and the figure caption has been revised to explicitly indicate that panels (a,b), (c,d), and (e,f) correspond to the FW, OW, and SW cases, respectively.

---

## Author Comment (AC2)

**Response to Reviewer 1**

**Title:** Implementation of a sigma coordinate system in PALM-Sigma v1.0 (based on PALM v21.10) for LES study of the marine atmospheric boundary layer

**Manuscript number:** egusphere-2025-4390

We take this opportunity to thank the editor and reviewers of our paper for their kind collaboration to the improvement of this manuscript. We have taken into account all the concerns raised and we have made suggested modifications, marked by yellow background in the revised manuscript.

**Comments and responses**

**Reviewer #1**

**General comments:**

**Synopsis:** The authors Xu Ning and Mostafa Bakhoday-Paskyabi report in their manuscript entitled "Implementation of a sigma coordinate system in PALM-Sigma v1.0 (based on PALM v21.10) for LES study of the marine atmospheric boundary layer" on the development of a large-eddy-simulation (LES) code based on the PALM LES framework, incorporating a modified vertical coordinate that accounts for the actual, instantaneous position of the atmospheric lower boundary. The authors use their code to investigate the interaction of the wave-induced effect between surface waves and the marine atmospheric boundary layer. Their results clearly demonstrate that the modelling approach commonly employed in atmospheric flow models—representing the ocean's influence on the atmosphere solely through a roughness length parameterized as a function of wave characteristics such as significant wave height and wave period—constitutes a strong simplification, especially in situations where the wave field is not in equilibrium with the wind field. For example, the authors show that the mismatch between wave direction and wind direction also influences properties of the marine atmospheric boundary layer above the wave-affected layer. When wind and waves are opposed, turbulence throughout the entire marine atmospheric boundary layer is enhanced compared to the case where wind and waves are aligned.

**Evaluation:** I would like to thank the authors for what I consider to be an excellent piece of work. The manuscript is both very well structured and very well written. I have only minor comments on the manuscript, and therefore I recommend its acceptance for publication in EGUsphere after minor revisions. I'll ask the authors to take the comments below into account when revising the manuscript.

**Response:** We thank the reviewer for the positive evaluation of our work and for the constructive suggestions. All comments have been carefully considered, and detailed responses are provided below.

**Major Comments:**

1. Page 1, line 24/25: "The latest high-performance computing (HPC) equipment enables numerical simulations with a magnitude of grid points up to 10^10 (Kröniger et al., 2018)" I suggest to find a more recently published paper as a reference for the number of grid points that can be handled by the latest generation of HPC clusters. I think it is slightly contradictory if you speak of latest HPC equipment (which should be from 2025), but refer to a paper from 2018. If no more recent publication is available I suggest to modify the sentence, e.g. as follows: "Already in 2018 high-performance computing (HPC) equipment enabled numerical simulations with a magnitude of up to 10^10 grid points (Kröniger et al., 2018)."

**Response:** We thank the reviewer for this helpful suggestion. To avoid a potential inconsistency between the wording and the publication year of the reference, we have revised the sentence as suggested to emphasize the demonstrated capabilities of HPC systems already in 2018, while retaining the original reference.

2. Page 2, line 27: Please correct "In numerical modeling of the marine atmospheric boundary layer flows" to "In numerical modeling of marine atmospheric boundary layer flows."

**Response:** We thank the reviewer for this correction. The text has been revised accordingly.

3. Page 3, line 68: Please change "The current work further develop PALM …" to "The current work further develops PALM …."

**Response:** The text has been revised accordingly.

4. Page 8, equation 23: The values of the weights applied in the Runge-Kutta scheme should be specified.

**Response:** We thank the reviewer for pointing this out. The values of the Runge–Kutta coefficients used in Eq. (23) have now been explicitly specified in the manuscript immediately following the equation.

5. Page 12, line 246: To apply the grid stretching already inside the atmospheric boundary layer is rather uncommon at least in work that has been done with the LES code PALM. Typically, the grid stretching would be applied above the atmospheric boundary-layer where turbulent processes do not play a major role any longer. It would be an extremely interesting result if the authors could show that applying the grid stretching already inside the boundary layer does not change the results significantly as it would be of importance for the computing resources required to carry out LES runs. Therefore, I ask the authors to extend their study by one extra simulation that applies a uniform distance in z-direction inside the boundary layer.

**Response:** We thank the reviewer for this valuable suggestion. Following this recommendation, we performed additional simulations using a uniform vertical grid spacing within the atmospheric boundary layer for both the wind-following (FW) and wind-opposing (OW) wave cases, while keeping all other model settings unchanged. The results are presented in appendix B with an additional figure. The comparison shows that the mean velocity profiles are virtually unaffected by the choice of vertical grid spacing. For the OW case, a moderate reduction of near-surface turbulent kinetic energy is observed when using a uniform grid, with a maximum absolute difference of approximately 0.08 m² s⁻², which gradually diminishes with height. This behavior suggests that turbulence statistics under wind-opposing wave conditions may be more sensitive to vertical resolution and therefore merit further investigation. Nevertheless, we believe that the observed differences do not significantly alter the overall flow structure nor the main conclusions of the present study.

6. Page 12, line 247: The authors report that their model domain has a size of 1200 m x 1200 x 850 m. The initial height of the bottom boundary of the inversion layer chosen by the authors is 600 m. I'm wondering whether the size of the model domain is actually large enough. I ask the authors to add a statement whether they checked the sensitivity of their results on the size of the model domain.

**Response:** We thank the reviewer for raising this important point. The choice of the model domain size and inversion layer height in this study was guided by both previous LES studies and physical scale considerations relevant to marine atmospheric boundary layer flows.

First, our domain configuration is consistent with earlier LES studies of wind-wave interaction. In particular, Sullivan et al. (2008) employed a domain size of 1200 m × 1200 m × 800 m under geostrophic wind forcing and wave conditions comparable to those used in the present study, but with a substantially shallower initial boundary layer depth of approximately 400 m, which is 200 m lower than that in our setup. In addition, Jiang et al. (2016) demonstrated that wave boundary layer characteristics and surface fluxes are relatively insensitive to the choice of inversion-layer height, supporting the adequacy of the present vertical domain extent.

Beyond literature precedent, the adequacy of the chosen domain size is further supported by the scales of turbulence. For a neutrally stratified atmospheric boundary layer, the dominant energy-containing turbulent eddies typically have horizontal length scales on the order of a few hundred meters, up to approximately the boundary layer depth (Moeng, C. H., 1984). The horizontal extent of 1200 m in the present simulations therefore exceeds the characteristic turbulent length scales by a factor of several, allowing for representing fully developed turbulence without artificial confinement effects.

Moreover, wave-induced modulations of the airflow are expected at length scales comparable to the dominant wavelength. In the present study, the wavelength is 100 m, which is more than one order of magnitude smaller than the horizontal domain size. This ensures that wave-induced flow structures and their interactions with atmospheric turbulence are well captured without being constrained by the lateral boundaries.

Taken together, the consistency with previous LES studies, the sufficient separation between domain size and dominant turbulent length scales, and between the domain size and the wave-induced flow structures provide confidence that the chosen model domain is sufficiently large for the objectives of the present study.

7. Page 12, line 249: The authors report that they have applied a total simulation time of 20 h. Is that length of the simulation run sufficient to get rid of the inertial oscillations that occur in simulations with PALM if the Coriolis force is switched on? g. Maas (2023) used a physical simulation time of 48 h to obtain a steady-state mean flow in his LES of an offshore flow at 55°N (Maas, 2023: From gigawatt to multi-gigawatt wind farms: wake effects, energy budgets and inertial gravity waves investigated by large-eddy simulations). In order to allow others to repeat their simulations the authors should provide also an information on which geographical latitude was assumed in their simulations. Moreover, information on the lateral boundary conditions chosen should be provided. Page 14, figure 3: I doubt slightly the value of showing a comparison of instantaneous velocity fields. The flow is anyhow different. We do not see the impact of background turbulence and turbulence created

by the different surface waves. I think to show statistical measures like the variance of velocity components makes more sense.

**Response:** We thank the reviewer for this important comment. In our simulations, the geographical latitude is set to 54° N, which is now explicitly stated in the revised manuscript and is close to the latitude used by Maas (2023).

To assess the spin-up behavior and the influence of inertial oscillations, we analyzed the temporal evolution of the horizontally averaged streamwise velocity profiles over the full 20 h simulation period. The results are shown in a new figure included in the Appendix A. As illustrated, pronounced inertial oscillations are present during the initial half inertial period (approximately 7 h). Subsequently, the mean velocity profiles exhibit a clear convergence toward a quasi-steady state, with only weak residual oscillations remaining toward the end of the simulations. The residual oscillations during the final hour, over which the statistical analyses presented in the manuscript are performed, are small compared to the early spin-up phase and do not affect the vertical structure of the mean flow. While longer integration times would further reduce the remaining inertial signal, the present results indicate that the flow has largely adjusted within the simulated period. The chosen simulation length of 20 h is therefore considered sufficient for the objectives of this study and represents a reasonable compromise given the available computational resources.

We have added a statement specifying the cyclic boundary conditions employed at the lateral planes in the simulation setup section.

We have revised Fig. 3 by complementing the instantaneous velocity fields with vertical profiles of the variances of u and w components overlaid on each subplot. These variance profiles provide a quantitative comparison of turbulence strength under different wave conditions and allow a clear distinction between background turbulence and wave-induced effects.

8. Page 15, figure 4: The caption of figure 4 lacks an information which cases are actually shown in the figure.

**Response:** We thank the reviewer for this comment. Figure 4 has been revised by adding subplot labels and by updating the figure caption to explicitly indicate which cases are shown.

9. General comment: I encourage the authors to mention also other possible applications of their modified PALM code such as flow in complex terrain in the outlook section of their manuscript.

**Response:** We thank the reviewer for this constructive suggestion. The conclusion has been revised to explicitly mention additional potential applications of PALM-Sigma beyond wind–wave interaction such as flow over complex terrain.

**References**

Sullivan, P. P., Edson, J. B., Hristov, T., & McWilliams, J. C. (2008). Large-eddy simulations and observations of atmospheric marine boundary layers above nonequilibrium surface waves. *Journal of the Atmospheric Sciences*, *65*(4), 1225-1245.

Jiang, Q., Sullivan, P., Wang, S., Doyle, J., & Vincent, L. (2016). Impact of swell on air–sea momentum flux and marine boundary layer under low-wind conditions. *Journal of the Atmospheric Sciences*, *73*(7), 2683-2697.

Moeng, C. H. (1984). A large-eddy-simulation model for the study of planetary boundary-layer turbulence. *Journal of Atmospheric Sciences*, *41*(13), 2052-2062.